# Glucocorticoid Receptor Regulates and Interacts with LEDGF/p75 to Promote Docetaxel Resistance in Prostate Cancer Cells

**DOI:** 10.3390/cells12162046

**Published:** 2023-08-11

**Authors:** Evelyn S. Sanchez-Hernandez, Pedro T. Ochoa, Tise Suzuki, Greisha L. Ortiz-Hernandez, Juli J. Unternaehrer, Hossam R. Alkashgari, Carlos J. Diaz Osterman, Shannalee R. Martinez, Zhong Chen, Isaac Kremsky, Charles Wang, Carlos A. Casiano

**Affiliations:** 1Center for Health Disparities and Molecular Medicine, Loma Linda University School of Medicine, Loma Linda, CA 92350, USA; esanchezhernandez@students.llu.edu (E.S.S.-H.); tsuzuki@students.llu.edu (T.S.); gortizhernandez@coh.org (G.L.O.-H.); junternaehrer@llu.edu (J.J.U.); halkashgari@students.llu.edu (H.R.A.); 2Department of Basic Sciences, Loma Linda University School of Medicine, Loma Linda, CA 92350, USA; zchen@llu.edu (Z.C.); ikremsky@llu.edu (I.K.); chwang@llu.edu (C.W.); 3Department of Physiology, College of Medicine, University of Jeddah, Jeddah 23890, Saudi Arabia; 4Department of Basic Sciences, Ponce Health Sciences University, Ponce, PR 00716, USA; cjdiaz@psm.edu (C.J.D.O.); shamartinez@psm.edu (S.R.M.); 5Center for Genomics, Loma Linda University School of Medicine, Loma Linda, CA 92350, USA; 6Rheumatology Division, Department of Medicine, Loma Linda University School of Medicine, Loma Linda, CA 92350, USA

**Keywords:** autoantibodies, chemoresistance, docetaxel, exicorilant, glucocorticoid receptor, LEDGF/p75, prostate cancer, relacorilant

## Abstract

Patients with advanced prostate cancer (PCa) invariably develop resistance to anti-androgen therapy and taxane-based chemotherapy. Glucocorticoid receptor (GR) has been implicated in PCa therapy resistance; however, the mechanisms underlying GR-mediated chemoresistance remain unclear. Lens epithelium-derived growth factor p75 (LEDGF/p75, also known as PSIP1 and DFS70) is a glucocorticoid-induced transcription co-activator implicated in cancer chemoresistance. We investigated the contribution of the GR–LEDGF/p75 axis to docetaxel (DTX)-resistance in PCa cells. GR silencing in DTX-sensitive and -resistant PCa cells decreased LEDGF/p75 expression, and GR upregulation in enzalutamide-resistant cells correlated with increased LEDGF/p75 expression. ChIP-sequencing revealed GR binding sites in the LEDGF/p75 promoter. STRING protein–protein interaction analysis indicated that GR and LEDGF/p75 belong to the same transcriptional network, and immunochemical studies demonstrated their co-immunoprecipitation and co-localization in DTX-resistant cells. The GR modulators exicorilant and relacorilant increased the sensitivity of chemoresistant PCa cells to DTX-induced cell death, and this effect was more pronounced upon LEDGF/p75 silencing. RNA-sequencing of DTX-resistant cells with GR or LEDGF/p75 knockdown revealed a transcriptomic overlap targeting signaling pathways associated with cell survival and proliferation, cancer, and therapy resistance. These studies implicate the GR–LEDGF/p75 axis in PCa therapy resistance and provide a pre-clinical rationale for developing novel therapeutic strategies for advanced PCa.

## 1. Introduction

Prostate cancer (PCa) is the second most common male cancer and the fifth most common cause of cancer-related death among men worldwide [1,2]. Although localized PCa has an excellent prognosis, the 5-year survival rate for patients with metastatic disease can be as low as 30% [3]. Androgen deprivation therapy (ADT) is initially effective until metastatic castration-resistant PCa (mCRPC), the terminal disease state, develops [4,5]. The treatment options for mCRPC have been evolving over the past few years, with combinatorial or sequential therapy using ADT, androgen receptor signaling inhibition (ARSI, using enzalutamide, apalutamide, darolutamide, and abiraterone acetate), and taxane chemotherapy with docetaxel (DTX) and cabazitaxel (CBZ) showing survival benefits and becoming the mainstays for patients with mCRPC [6]. Unfortunately, resistance to these therapies ultimately develops, leading to patient death. Understanding the mechanisms underlying this resistance is critical for the development of more effective therapies for patients with mCRPC. 

Corticosteroids may influence PCa cell proliferation in a context-dependent manner by activating mutated (L702H) androgen receptor (AR), thus promoting mCRPC progression and therapy resistance [7,8,9]. The upregulation of AR splice variants such as AR-V7 has also been shown to contribute to resistance to both ARSI and chemotherapy in mCRPC [10,11]. Further, mCRPC cells treated with enzalutamide and abiraterone upregulate glucocorticoid receptor (GR) expression, and GR signaling leads to the transactivation of shared AR target genes, thus bypassing AR blockade [12,13,14,15]. Clinical evidence supports the presence of AR L702H mutation and enhanced GR expression in patients with PCa who acquired resistance to ARSI [12,14,16]. Although the role of GR signaling in ARSI resistance is well established [12,13,14,17,18,19,20,21,22], there is limited evidence linking GR signaling to PCa chemoresistance [20,23,24,25]. Emerging evidence supports the concept of therapy cross-resistance in PCa, wherein a particular therapy may lead to resistance to a subsequent therapy [26]. The mechanistic contribution of GR signaling to PCa therapy cross-resistance is unclear.

GR signaling may contribute to PCa chemoresistance via nuclear protein–protein interactions between GR and β-catenin [25] and the glucocorticoid-mediated upregulation of oncoproteins such as lens epithelium derived growth factor p75 (LEDGF/p75) and clusterin [24]. LEDGF/p75 is overexpressed in several human cancers, including PCa, and promotes tumor aggressive properties such as enhanced cancer cell proliferation, migration, clonogenic growth, tumorsphere formation, DNA repair, angiogenesis, and resistance to various chemotherapeutic agents [27,28,29,30,31,32,33,34,35,36,37,38,39]. Acting as a transcription co-activator, LEDGF/p75 promotes cell survival under diverse environmental stressors, including radiation, heat, serum starvation, oxidative stress, and cytotoxic agents, by contributing to the upregulation of antioxidant and stress survival genes [40,41,42,43,44,45,46,47]. LEDGF/p75 appears to work in concert with the hepatoma-derived growth factor (HDGF)-related protein 2 (HRP2, HDGF2, or HDGFL2). Both proteins belong to the HDGF family, are involved in leukemic cell survival [48], and are critical factors to facilitate RNA polymerase II transcription by relieving the nucleosome-mediated barrier for transcription elongation [49]. In addition, both proteins have an N-terminal PWWP domain involved in active chromatin reading [48,49] and a C-terminal integrase binding domain (IBD) that is key for HIV integration into the host chromatin [48,50,51] and for protein–protein interactions with multiple oncogenic transcription factors [52,53]. 

In a previous study, we showed that LEDGF/p75 and members of its transcriptional network such as Menin, JPO2, and HRP2 contribute to the survival, clonogenicity, and tumorsphere formation capacity of DTX-resistant PCa cells [38]. In the present study, we evaluated the potential regulation of LEDGF/p75 by GR and investigated the inhibitory activity of selective GR modulators (SGRMs) in DTX-resistant PCa cells. We provide evidence that GR silencing decreases LEDGF/p75 but not HRP2 protein expression in a panel of PCa cell lines, and that like GR, LEDGF/p75 is also upregulated in LNCaP enzalutamide-resistant (LNCaP-ENZR) cells. In addition, we show that GR and LEDGF/p75 interact endogenously and co-localize in the nuclei of DTX-resistant cells and that their co-targeting increases the response of chemoresistant PCa cells to DTX. Further, the transcriptomic profiling of chemoresistant PCa cells depleted of GR or LEDGF/p75 revealed differential expression of unique and overlapping target genes and pathways associated with cancer cell survival and therapy resistance. Our results provide novel insights into the contribution of the GR–LEDGF/p75 transcriptional axis to PCa chemoresistance. 

## 2. Materials and Methods

### 2.1. Cell Culture

PCa cell lines PC3 (CRL-1435), DU145 (HTB-81), 22Rv1 (CRL-2505), MDA-PCa-2b (CRL-2422), and LNCaP (CRL-1740) were purchased from the American Type Culture Collection (ATCC, Manassas, VA, USA). MDA-PCa-2b cells were cultured in F-12K medium (Corning, Glendale, AZ, USA, Cat# 10-025-CVR) supplemented with 20% fetal bovine serum (FBS, Genesee Scientific, San Diego, CA, USA, Cat# 25-514), 1% Penicillin/Streptomycin (Corning, Glendale, AZ, USA, Cat# 30-002-CI), 25 ng/mL cholera toxin (Sigma-Aldrich, St. Louis, MO, USA, Cat# 8052), 10 ng/mL human epidermal growth factor (Sigma-Aldrich, St. Louis, MO, USA, Cat# E9644), 5 μg/mL insulin (Sigma-Aldrich, St. Louis, MO, USA, Cat# I0516), 100 pg/mL hydrocortisone (Sigma-Aldrich, St. Louis, MO, USA, Cat# H0135), 5.8 ng/mL selenous acid (Sigma-Aldrich, St. Louis, MO, USA, Cat# 229857), and 700 ng/mL *O*-phosphorylethanolamine (Sigma-Aldrich, St. Louis, MO, USA, Cat# P0503). PC3, DU145, 22Rv1, and LNCaP cells were cultured in RPMI-1640 medium (Genesee Scientific, San Diego, CA, USA, Cat# 25-506), supplemented with 10% FBS, 1% Penicillin/Streptomycin (Corning, Glendale, AZ, USA, Cat# 30-002-CI), and Normocin 1G (Fisher Scientific, Pittsburgh, PA, USA, Cat# NC9390718). DTX-resistant PC3 (PC3-DR), DU145 (DU145-DR), and 22Rv1 (22Rv1-DR) cell lines were developed as previously described [25,29] and cultured in medium containing 10 nM DTX (LC Laboratories, Woburn, MA, USA, Cat# D-1000). For the development of LNCaP-ENZR cells, an androgen-depleted LNCaP subline was first generated through gradual replacement of FBS with charcoal-stripped FBS (CS-FBS) in culture. Cells resistant to androgen depletion were maintained in medium supplemented with 10% CS-FBS and subsequently exposed to incrementally increasing concentrations of enzalutamide to a final concentration of 50 μM in culture. Surviving cells were expanded and maintained at this concentration. Cell lines were routinely tested for mycoplasma contamination using the MycoAlert Plus assay (Lonza, Walkersville, MD, USA, Cat# LT07-218). Short tandem repeat (STR) profiling from ATCC (Cat# ATCC-135-XV) was utilized to authenticate all the cell lines.

### 2.2. Small Interfering RNA (siRNA)-Mediated Knockdown

Cells were transiently transfected for 72 h with scrambled (SCR) negative control (Integrated DNA Technologies, Coralville, IA, USA, Cat# 51-01-19-09), tri-silencer siRNA targeting GR (5′-AGAAUGACCUACAUCAAAGAGCUAG, 5′-GGAUACUAUACAAG CAGAACUGAGG, and 5′-GGAGAUCAUAUAGACAA UCAAGUGC), or siRNA targeting LEDGF/p75 (5′-AGACAGCAUGAGGAAGCGAUU). siRNA transfections (25 nM or 50 nM) were performed as described [25,38] and confirmed by immunoblotting.

### 2.3. Quantitative Reverse Transcription PCR (RT-qPCR)

Total RNA was extracted from cell lines using the RNeasy plus mini kit (Qiagen, Redwood City, CA, USA, Cat# 74134). RNA (0.5 μg) and reverse-transcribed into cDNA using iScript cDNA synthesis kit (Bio-Rad, Hercules, CA, USA, Cat# 1708891). Primer sequences were commercially synthesized by Integrative DNA Technologies (IDT). The forward sequence for LEDGF/p75 (5′ to 3′) was TGCTTTTCCAGACATGGTTGT and reverse sequence (5′ to 3′) was CCCACAAACAGTGAAAAGACAG. The forward sequence for GR was TCTGAACTTCCCTGGTCGAA and reverse sequence was GTGGTCCTGTTGTTGCTGTT. The forward sequence for GAPDH was CTCCTCCACCTTTGACGCTG and reverse sequence was TCCTCTTGTGCTCTTGCTGG. Quantitative PCR was performed on the MyiQ real-time PCR and CFX96 Touch Real-Time PCR (Bio-Rad) detection system using iQ SYBR Green Supermix (Bio-Rad, Hercules, CA, USA, Cat # 170-8882) according to the manufacturer’s instructions. The cycling conditions were 95 °C for 15 min, 95 °C for 15 s, and 60 °C for 60 s, for 35 cycles, followed by melt analysis from 60 to 95 °C. GAPDH mRNA levels were used for normalization. Data were analyzed from 2 independent biological replicates performed experimentally in triplicates.

### 2.4. Immunoblotting

Whole cell lysates were prepared as described [25] and 25 μg of total proteins were loaded into individual lanes of 4–12% bis-tris SDS-polyacrylamide gels (Fisher Scientific, Pittsburgh, PA, USA). After electrophoresis, proteins were then transferred to PVDF membranes using Invitrogen iBlot^TM^ 2 PVDF Mini Stacks (Fisher Scientific, Pittsburgh, PA, USA, Cat# IB24002). Membranes were blocked in 5% dry milk in tris-buffered saline buffer (TBS-T) containing 0.2% tween-20, 20 mM Tris–HCl, pH 7.6, and 140 mM NaCl. Membranes were probed with corresponding primary antibodies overnight at 4 °C. Primary antibodies included rabbit antibodies to LEDGF/p75 (Bethyl Laboratories/Fortis Life Sciences, Montgomery, TX, USA, Cat# A300-848A), GR (Cell Signaling Technology, Danvers, MA, USA, Cat# 12041S, clone D6H2L), GAPDH (Cell Signaling Technology, Danvers, MA, USA, Cat# 2118S, clone 14C10), HRP2 (Bethyl Laboratories/Fortis Life Sciences, Montgomery, TX, USA, Cat# A304-314A), and H3K36me2 (Cell Signaling Technology, Danvers, MA, USA, Cat# 2091T, clone C75H12) or mouse anti-Bcl-2 (Santa Cruz Biotechnology, Inc, Dallas, TX, USA, Cat# sc-509). Human sera containing autoantibodies specific for DNA topoisomerase I (TOP1/Scl-70) and poly (ADP-ribose) polymerase (PARP) were from the Casiano Lab autoimmune serum collection. Incubation of membranes with primary antibodies was followed by multiple washes with TBS-T and incubation with HRP-linked anti-rabbit and anti-mouse IgG secondary antibody (Cell Signaling Technology, Danvers, MA, USA, Cat# 7074S and 7076S) and HRP-linked goat anti-human (Invitrogen-ThermoFisher Scientific, Waltham, MA, USA, Cat# A18847). Enhanced chemiluminescence (Fisher Scientific, Pittsburgh, PA, USA, Cat# PI34580) was used to detect immunoreactive bands in autoradiography film (Midwest Scientific, Fenton, MO, USA, Cat# XC6A2). Protein bands were quantified using ImageJ software (National Institutes of Health, Bethesda, MD, USA, Fiji Version 1.44a), and relative expression was calculated after normalizing to glyceraldehyde-3-phosphate dehydrogenase (GAPDH).

### 2.5. Immunoprecipitation

Endogenous LEDGF/p75 immunoprecipitation (IP) was performed from whole cell lysates using an immunoprecipitation kit following the manufacturer’s protocol (Abcam, Waltham, MA, USA, Cat# ab206996). Briefly, cells were grown in 100 mm tissue culture-treated dishes (Genesee Scientific, San Diego, CA, USA, Cat# 25-202) until they reached 80–90% confluency. Cell viability was assessed prior to IP and the procedure was performed if the viability exceeded 90%. Cells were washed with ice cold DPBS, gently scraped in non-denaturing lysis buffer plus protease inhibitor cocktail (PIC) provided by the kit, and collected in chilled 1.5 mL microcentrifuge tubes. Tubes were mixed on a rotator mixer for 30 min at 4 °C and then centrifuged at 10,000× *g* for 10 min at 4 °C. Supernatant soluble proteins (500 μg) were gently mixed overnight on a rotator mixer at 4 °C with pre-washed protein A/G Sepharose beads in wash buffer, high titer monospecific anti-LEDGF/p75 human autoantibodies (1:100), or irrelevant normal human serum (NHS) that lacks reactivity against LEDGF/p75 as negative control. In some experiments, the lysates were treated with DNase I (Qiagen, Redwood City, CA, USA, Cat# 79254) for 15 min to digest chromatin prior to the addition of protein A/G Sepharose beads with immunoprecipitating antibodies. The specificity of the anti-LEDGF/p75 autoantibodies used for IP was established in a previous study [38]. The beads with the antigen–antibody complexes were centrifuged at 2000× *g* for 2 min at 4 °C and washed three times with wash buffer. Immunoprecipitated proteins were eluted with 4x lithium dodecyl sulfate (LDS) buffer (Fisher Scientific, Pittsburgh, PA, USA, Cat# NP0007) containing 1% β-mercaptoethanol (Sigma-Aldrich, St. Louis, MO, USA, Cat# M-6250), and this was followed by boiling for 5 min. Samples were centrifuged at 12,000 rpm for 3 min at 4 °C prior to SDS-PAGE analysis.

### 2.6. Confocal Microscopy

Cells were seeded on glass coverslips placed inside 6-well plates (Genesee Scientific, San Diego, CA, USA, Cat# 25-105) in complete RPMI media, allowed to adhere for 24 h, and then incubated for 12 h in medium supplemented with 10% CS-FBS (Fisher Scientific, Pittsburgh, PA, USA, Cat# 12676-029). Cells were treated with 100 nM dexamethasone (Sigma-Aldrich, St. Louis, MO, USA, Cat# D4902) for 30 min followed by fixation with 4% paraformaldehyde (Electron Microscopy Sciences, Hatfield, PA, USA, Cat# 15712) and permeabilization with 0.2% Triton X-100 (Fisher Scientific, Pittsburgh, PA, USA, Cat# BP151-100). Cells were then incubated with blocking buffer (10% Triton-X100, 12.5% BSA, 0.5% Tween-20 in DPBS) for 1 h and then with high titer, monospecific anti-LEDGF/p75 human autoantibodies together with rabbit anti-GR commercial antibody (1:200 antibody dilution) for 2 h at room temperature. Cells were washed with DPBS and then incubated with goat anti-human IgG (H + L) FITC (ThermoFisher Scientific, Waltham, MA, USA, Cat# 62-711) together with goat anti-rabbit IgG (H + L) Rhodamine (Millipore Sigma, St. Louis, MO, USA, Cat# AP124R) diluted in blocking solution (1:50 antibody dilution) for 1 h. VECTASHIELD Antifade Mounting Medium with DAPI (Vector Laboratories, Newark, CA, USA, Cat# H-1200-10) was used and images were acquired at 63X magnification in a Zeiss LSM-710-NLO confocal microscope.

### 2.7. Cell Viability and Apoptosis Assays

Cells were seeded in triplicate wells in 96-well plates (Genesee Scientific, San Diego, CA, USA, Cat# 25-109), allowed to adhere for 24 h, and treated with dimethylsulfoxide (DMSO, Fisher Scientific, Pittsburgh, PA, USA, Cat# PI20688) as vehicle control or with increasing concentrations of DTX (0–10,000 nM) for 72 h. Following treatments, 25μL of 5 mg/mL MTT solution prepared in DPBS was added into each well and plates were incubated for 2 h at 37 °C. Plates were then centrifuged for 5 min at 1500 rpm, supernatants were discarded, and formazan (MTT metabolite) within the cells was subsequently solubilized with 100μL DMSO per well. Absorbance was measured at 490 nm using a SpectraMax spectrophotometer (Molecular Devices LLC, San Jose, CA, USA). Absorbance readings were normalized to vehicle-treated values, and the IC_50_ was determined in DTX-sensitive and -resistant PCa cells. To investigate the cytotoxicity of the SGRMs exicorilant (EXI, CORT125281, Corcept Therapeutics, Menlo Park, CA, USA) and relacorilant (RELA, CORT125134, Corcept Therapeutics, Menlo Park, CA, USA), cells were treated for 72 h with increasing concentrations of EXI or RELA (0–10,000 nM) in the presence or absence of 10 nM DTX. To evaluate whether these SGRMs increased cellular sensitivity to DTX, DTX-resistant PCa cells were treated with 1 μM, 5 μM, or 10 μM of EXI or RELA in combination with increasing DTX concentrations (0–10,000 nM). Following MTT assay, the IC_50_ was then calculated to determine resensitization to DTX.

For apoptosis assays, PC3-DR and DU145-DR cells were transfected for 72 h with SCR negative control and siRNA targeting GR. Supernatant from each condition was collected prior to detaching the adherent monolayer of cells with diluted trypsin and harvesting. The combined floating and attached harvested cells for each condition were used for analysis, and samples were kept on ice. Annexin V/7-AAD staining was performed using the Annexin V Apoptosis Detection Kit eFluor^TM^ 450 (eBioscience-Thermo Fisher Scientific, Waltham, MA, USA, Cat#88800672) according to the manufacturer’s protocol. Fluorescence was measured using a Miltenyi Biotec MACSQuant Analyzer 10 Flow Cytometer (Miltenyi Biotec, Auburn, CA, USA). The percentage of apoptotic cells (Annexin V positive) was determined using FlowJo software 9.9.6 (FlowJo, Ashland, OR, USA).

### 2.8. Clonogenic Assays

PC3-DR cells were seeded in 6-well plates (1000 cells/well) and allowed to adhere for 24 h. Cells were then treated with vehicle (DMSO), EXI, or RELA (1, 5, 10 μM) in the presence or absence of 10 nM DTX. To assess the effects of targeting LEDGF/p75 or GR on clonogenicity, PC3-DR cells transfected with SCR negative control or siLEDGF/p75 were plated and, 24 h later, treated with vehicle (DMSO) or 1 μM EXI or RELA. Colonies were grown for 10 days, and surviving colonies were fixed with methanol:acetic acid (3:1) solution for 5 min and washed with DPBS before and after fixation. Colonies were stained with 0.5% crystal violet (Sigma-Aldrich, St. Louis, MO, USA, Cat# C0775) and dissolved in methanol (Fisher Scientific, Pittsburgh, PA, USA, Cat# A412-4) for 20 min. Crystal violet solution was removed and this was followed by gentle washing with water. Colonies were airdried overnight and imaging was performed with a ChemiDoc^TM^ MP Imaging System (BioRad, Hercules, CA, USA). Colony quantification was performed with ImageJ software’s automated colony counting feature using identical parameters for all wells.

### 2.9. ChIP-Sequencing

ChIP-Atlas was used for the identification of publicly available ChIP datasets for *NR3C1,* the gene encoding GR [54,55]. The search parameters were set as follows: within the peak browser, the hg38 genome assembly/index was selected for *Homo sapiens*; for experiment type, “ChIP: TF and others” was chosen. All cell types were considered, the threshold for peak calling was set to Q < 1 × 10^−5^, and *NR3C1* was selected as the antigen. Integrative Genomics Viewer (IGV) was used to identify peaks near the transcription start site of *PSIP1,* the gene encoding LEDGF/p75 [56]. ChIP-Atlas bigWig tracks were also visualized in the UCSC human genome browser (GRCh37). Based on peak, antibody, and experimental setup quality (number of replicates, treatments, and controls), the GSE30623 and GSE39879 PCa datasets were selected for further analysis [57,58]. The following PCa samples were analyzed: SRR309201 (LNCaP-1F5; GR antibody; no treatment), SRR531806 and SRR531815 (LNCaP-1F5; GR antibody; dexamethasone-treated), SRR531816 (LNCaP-1F5; GR antibody; dexamethasone- and dihydrotestosterone-treated), and SRR531811 (VCaP; GR antibody; dexamethasone-treated). In addition, the acute lymphocytic leukemia (ALL) GSE175482 dataset was also selected for further analysis [59]. This dataset was derived from ChIP-seq analysis of the ALL cell lines 607 and Nalm6 exposed for 24 h to prednisolone (glucocorticoid used for GR activation; 10μM for 697 and 5μM for Nalm6 cells). For the ChIP-seq analysis, sample files were obtained from GEO with the SRA Toolkit [60]. Sample quality was determined with FastQC v0.11.9 and MultiQC v1.14 [61,62]. Reads were trimmed with Trimmomatic v0.39 [63], aligned to the human reference assembly, hg38, with bowtie2 [64], and converted to binary and indexed with SAMtools [65]. Downstream peak calling was done with both Homer v4.11 and MACS2 [66,67]. Motif enrichment analysis and peak annotation were also obtained with Homer v4.11. ChIPseeker and the UCSC genome browser were used for peak visualization [68,69,70,71,72,73]. UCSC genome browser session can be explored with the following link: https://genome.ucsc.edu/s/tsuzuki/ALL_PC_hg38, accessed on 31 May 2023. Red highlights indicate potential binding sites for GR/NR3C1 as identified with JASPAR 2022 [https://doi.org/10.1093/nar/gkab1113, accessed on 31 May 2023].

### 2.10. RNA-Sequencing

Total RNA was extracted from PC3-DR and DU145-DR cells transfected with SCR negative control, siLEDGF/p75, or siGR using the miRNeasy Mini Kit (Qiagen, Redwood City, CA, USA, Cat# 217004). All RNA samples were derived from three independent experiments. RNA-seq library construction and sequencing were performed at Loma Linda University Center for Genomics. RNA-seq library was constructed using the NuQuant Universal RNA-Seq library preparation kit (Tecan, Männedorf, Switzerland) following the manufacturer’s protocol. Briefly, 100 ng of total RNA was used as input. After first and second strand of cDNA synthesis, end-repair, adaptor index ligation, and strand selection were conducted, barcodes with unique indices were used for each sample for multiplexing. Ribosomal RNA depletion was performed using custom InDA-C primer mixture for human samples. Libraries were amplified for 13 cycles (Mastercycler^®^ pro, Eppendorf, Hamburg, Germany) and purified with Agencourt XP beads (Beckman Coulter, Indianapolis, IN). Purified libraries were quantified using the Qubit dsDNA HS Kit on a Qubit 4.0 Fluorometer (Life Technologies, Carlsbad, CA, USA). Quality and peak size were determined with the D1000 ScreenTape on Agilent 2200 TapeStation (Agilent Technologies, Santa Clara, CA, USA). RNA-seq libraries were sequenced on an Illumina NextSeq 550 with high output kit (Illumina, Inc., San Diego, CA, USA) at single-end 84 bp. Illumina RTA v2.4.11 software was used for basecalling and bcl2fastq v2.17.14 was used for generating fastq files.

### 2.11. Bioinformatics

RNA-seq data analysis and visualization were performed using the following pipeline: quality control check (FastQC), trimming process (TrimGalore), alignment (Tophat2), reads quantification (Cufflinks), and quantification of differentially expressed genes (Cuffdiff) [74,75,76,77]. Raw fastq files were first trimmed using TrimGalore and the trimmed reads were aligned to the human reference genome (NCBI GRCh38) using TopHat V2.1.1. Once aligned, bam files were processed using cufflinks for gene quantification. Differentially expressed genes (DEGs) were determined using Cuffdiff with q < 0.05 and log fold change (log FC) > 1. Principal component analysis (PCA) and hierarchical clustering analysis of global genes for all cell lines were performed with “R” program (http://cran.r-project.org, accessed on 3 December 2022) and Partek Genomics Suite 6.6, respectively. GSEA (v3.0, Broad Institute), was performed to compare SCR control PC3-DR and DU145-DR samples with PC3-DR and DU145-DR samples containing individual GR or LEDGF/p75 depletion. In silico analysis of GR and LEDGF/p75 protein interaction networks was performed using the STRING platform (https://string-db.org/cgi/about, accessed on 11 May 2023). The search focused on GR, using the gene name *NR3C1*, and LEDGF/p75, using *PSIP1*, and was expanded to 20 shell interactors including known LEDGF/p75-IBD interacting partners.

### 2.12. Statistics

GraphPad Prism, version 8.2.1, was used for statistical analysis and generation of graphs. Differences between treatment groups were analyzed using unpaired *t* test. Mean +/− SEM was calculated from at least 3 independent experiments and statistical significance was determined at *p* < 0.05.

## 3. Results

### 3.1. GR Depletion Leads to Decreased LEDGF/p75 Protein Expression in Prostate Cancer Cells

Previously, we demonstrated that cortisol and the synthetic glucocorticoid dexamethasone induced the expression of LEDGF/p75 in selected PCa cell lines [24]. To further investigate the potential regulation of LEDGF/p75 by activated GR, we assessed the effects of GR silencing on LEDGF/p75 protein expression in a panel of DTX-sensitive and DTX-resistant PCa cell lines. siRNA-mediated GR knockdown (siGR) was confirmed in the DTX-sensitive cell lines PC3 (AR−/GR+), DU145 (AR−/GR+), 22Rv1 (AR+/GR+), and MDA-PCa-2b (AR+/GR+) by immunoblotting analysis (Figure 1A–D). This knockdown did not cause a significant increase in cell death in the DTX-resistant PC3-DR and DU145-DR cell lines (Appendix A), which express high levels of GR [23,25]. These findings are consistent with our previous observation that the pharmacological targeting of GR alone in these cell lines induces minimal cell death [25]. GR depletion resulted in a significant decrease of LEDGF/p75 protein levels in all four cell lines (Figure 1A–D). Similar results were obtained in the DTX-resistant cell lines PC3-DR, DU145-DR, and 22Rv1-DR (Figure 1E–G), implicating GR as a regulator of LEDGF/p75 expression in multiple PCa cell lines. 

We ruled out that the decreased LEDGF/p75 protein expression was due to cytotoxicity induced by the siGR transfections since, as mentioned above, cell viability was unaffected in the transfected cultures. Further, siGR transfection led to decreases in both GR and LEDGF/p75 mRNA expression (Appendix A), indicating that the observed decrease in protein expression had resulted from transcript suppression. 

Next, we explored whether GR depletion also influenced the expression of HRP2. LEDGF/p75 and HRP2 belong to the HDGF family of proteins, share high homology in their N-terminal PWWP domain, and are the only two members of this family that also have a C-terminal Integrase Binding Domain (IBD) (Figure 2A) [48,78,79]. Both proteins share multiple interacting partners and, given their structural and functional overlap, are considered paralogs that contribute to HIV integration, RNA polymerase II (RNAPII) transcription, and cell survival in leukemia and chemoresistant PCa cells [38,48,49,50,51].

We evaluated the protein expression of HRP2 in the same panel of DTX-sensitive and DTX-resistant PCa cell lines depleted of GR to determine whether, like LEDGF/p75, this protein is also regulated by GR. However, we observed no significant changes in HRP2 protein expression after GR silencing in all the PCa cell lines tested (Figure 2B–H). Of note, although there was a noticeable reduction in HRP2 expression in 22Rv1 and 22Rv1-DR cells with GR silencing, it did not achieve statistical significance (Figure 2D,H). While these immunoblots were repeated independently three times, we cannot rule out that with additional replicates, this reduction may attain significance. It is not clear, however, why GR would selectively influence HRP2 expression in 22Rv1 cell lines.

In related experiments, we explored whether there is an interdependency between the expression of GR and LEDGF/p75 in PCa cells by silencing the latter in the same panel of DTX-sensitive and DTX-resistant PCa cell lines and assessing GR protein expression levels. After confirming LEDGF/p75 depletion by immunoblotting, no statistically significant changes were observed in GR protein expression in any of the siLEDGF/p75 samples compared to the SCR controls (Appendix A). 

Taken together, these results suggest that GR regulates the protein expression of LEDGF/p75, but not that of HRP2, in DTX-sensitive and DTX-resistant PCa cells with the possible exception of the 22Rv1 cell line pair. By contrast, LEDGF/p75 does not appear to be required to maintain GR protein expression in these cells.

### 3.2. GR and LEDGF/p75 Are Upregulated in Enzalutamide-Resistant LNCaP Cells

Evidence from our group and others indicates that both GR and LEDGF/p75 are upregulated in DTX-resistant PCa cells compared to parental drug-sensitive cells [23,25,29,38]. GR expression is also upregulated in enzalutamide-resistant tumors in vivo and in biopsies from patients with metastatic PCa treated with enzalutamide [12,13]. In addition, the long-term treatment of PCa cells with ARSI (enzalutamide and abiraterone) leads to increased GR expression, which correlates with early biochemical relapse [14]. To determine whether GR upregulation correlates with increased LEDGF/p75 expression in enzalutamide-resistant cells, we assessed their protein expression levels in LNCaP (AR+/GR−) and LNCaP-ENZR cells. LNCaP-ENZR cells were derived from their parental LNCaP cells by incrementally exposing the latter to increasing concentrations of enzalutamide, with this being followed by the selection of surviving cells. We observed that the protein expression of both GR and LEDGF/p75 was significantly upregulated in LNCaP-ENZR cells when compared to parental LNCaP cells (Figure 3A,B). 

To determine whether enhanced LEDGF/p75 expression in LNCaP-ENZR cells was dependent on GR, we performed GR silencing in these cells. In agreement with our results from DTX-sensitive and DTX-resistant cells, LEDGF/p75 expression was significantly diminished after GR depletion (Figure 3C,D). These findings further confirmed the dependence of LEDGF/p75 protein expression on GR expression in PCa cells and implicated the GR–LEDGF/p75 axis in enzalutamide resistance. 

### 3.3. ChIP-Seq Analysis Reveals GR Binding Sites in the Promoter Region of LEDGF/p75 

GR exerts its transcriptional functions by binding to glucocorticoid response elements (GREs) in the promoter regions of its target genes and by interacting with transcription factors and co-regulators to regulate transcription. Putative GR binding sites were previously identified by our group in the gene encoding LEDGF/p75, *PSIP1* [24]; however, direct GR binding to this promoter has not been reported. The screening of publicly available ChIP-seq datasets for putative GR binding sites within the *PSIP1* promoter region identified several studies showing GR binding to this promoter. For further analysis, we selected ChIP-seq data sets from two independent studies focused on glucocorticoid-treated PCa cell lines (LNCaP-1F5 and VCaP) and two ALL cell lines (697 and Nalm6) [57,58,59]. Of note, the LNCaP-1F5 cell line was engineered to overexpress GR and is, to some extent, equivalent to our LNCaP-ENZR cell line, which naturally overexpresses GR after selection for enzalutamide resistance (Figure 3A,B). The VCaP cell line and the ALL cell lines endogenously express GR. In addition to its roles as an oncoprotein in PCa and other solid tumors, LEDGF/p75 has also been functionally implicated in leukemic gene rearrangements, transformation, and chemoresistance [31,37,48,80,81,82,83]. To determine whether GR (NR3C1) binding sites are present in the *PSIP1* promoter region in the PCa and ALL cell lines, the GR ChIP-seq datasets were analyzed following the workflows shown in Figure 4A and Appendix A.

The pie chart showing % feature distribution throughout the entire human genome indicates that GR binding sites predominantly occur in the intron, distal intergenic, and promoter regions for the PCa cell lines (comprising 46.99%, 29.94%, and 18.10%, respectively), and in the promoter, intron, and distal intergenic regions for the ALL cell lines (35.25%, 34.90%, and 23.88%) (Figure 4B and Appendix A).

To evaluate enrichment at the transcriptional start site (TSS), profile plots and heatmaps were generated showing GR binding density near all TSSs (Figure 4C and Appendix A). Binding motif analysis confirmed a GR binding site within 2000 bp from the TSS of *PSIP1* in chromosome 9p22.3 (the *PSIP1* gene location). The most statistically significant enriched GR binding motifs near the TSS of *PSIP1* for the PCa and ALL cell lines were identified (Figure 4D and Appendix A). Representative images of peaks demonstrating GR binding to the *PSIP1* promoter region in the PCa cell lines LNCaP-1F5 and VCaP (Figure 4E) and ALL cell lines 697 and Nalm6 (Appendix A) are shown. These results enrich our previous report of putative GREs in the promoter region of LEDGF/p75 [24] and strengthen our hypothesis that LEDGF/p75 is a target gene of GR.

### 3.4. LEDGF/p75 and GR Interact in DTX-Resistant PCa Cells

In a recent study, we demonstrated that LEDGF/p75 interacts with several members of its IBD protein interactome (e.g., Menin, MLL, JPO2, and PogZ) in DTX-resistant PCa cells and that some of these interactions are critical for maintaining cell survival, clonogenicity, and tumorsphere formation [38]. In addition, we observed that dexamethasone induces JPO2 nuclear translocation in PC3-DR and DU145-DR cells and that translocated JPO2 co-localizes in the nucleus with both GR and LEDGF/p75 [38], suggesting that these proteins belong to the same transcriptional network. To explore a possible interaction between GR and LEDGF/p75, we first performed a bioinformatic inquiry using the STRING platform, which provides valuable insights into protein–protein networks and their functional interactions (Figure 5). 

This protein interaction analysis revealed that GR and LEDGF/p75 belong to a large transcriptional network that also includes known direct LEDGF/p75 interactors such as Menin (MEN1), MLL (KMT2A), JPO2 (CDCA7L), H3K36me2/3 (HIST2H3D), IWS1, and POGZ, as well as likely indirect interactors such as GR (NR3C1), AR, β-catenin (CTNNB1), HRP2, MYC, BRD4, MED1, CREBBP, WDR5, and MAX among many others. Interestingly, DNA topoisomerase I (TOP1), which is not known to interact with either GR or LEDGF/p75, appeared outside the network. 

To determine experimentally whether LEDGF/p75 interacts with GR, we performed a co-immunoprecipitation analysis in DTX-resistant cells using a human serum previously characterized for the presence of monospecific high-titer autoantibodies to LEDGF/p75 [38]. These cells were grown in normal culture medium, which contains glucocorticoids. The immunoprecipitation of LEDGF/p75 by the human autoantibodies in whole cell lysates from DTX-resistant PC3-DR, DU145-DR, and 22Rv1-DR was confirmed by immunoblotting using commercially available rabbit antibodies to this protein (Figure 6A–C). GR and LEDGF/p75 co-immunoprecipitated (Figure 6A–C). H3K36me2, an active chromatin marker and interacting partner of LEDGF/p75 [38,53,84], was included as a positive control and was also co-immunoprecipitated by the LEDGF/p75 autoantibodies (Figure 6A–C). 

To confirm that these interactions are specific, we conducted co-immunoprecipitation with a normal human serum that does not contain anti-LEDGF/p75 autoantibodies [38], observing that this serum does not precipitate LEDGF/p75, GR, or H3K36me2. We also observed that topoisomerase 1 (TOP1), did not co-immunoprecipitate with these three proteins, consistent with the STRING analysis and our previous observation that this protein is not a component of the LEDGF/p75 interactome [38]. 

These results suggest that LEDGF/p75 and GR are part of a transcriptional complex in the active chromatin of DTX-resistant PCa cells. However, to rule out that the LEDGF/p75-GR interaction is dependent on the binding of both proteins to chromatin, we conducted a co-immunoprecipitation experiment in DU145-DR cells, using the LEDGF/p75 autoantibodies, in the presence and absence of DNase treatment. This treatment did not affect the co-immunoprecipitation of LEDGF/p75 and GR (Appendix A), suggesting that this interaction is mediated by protein–protein interactions rather than the binding of GR and LEDGF/p75 to chromatin. In addition, we performed co-immunoprecipitations in DTX-sensitive cells, using the LEDGF/p75 autoantibodies, but were unable to detect LEDGF/p75 precipitation (Appendix A). We encountered a similar challenge in our previous study, which led us to focus our analysis on the LEDGF/p75 interactome in DTX-resistant cells [38]. In that study, we showed that LEDGF/p75 expression levels in DTX-sensitive cells are decreased compared to their resistant counterparts, which could explain the low efficiency of LEDGF/p75 immunoprecipitation in these cells. 

In another recent study, our group also showed that GR and β-catenin co-immunoprecipitate in PCa cells and contribute to DTX resistance [25]. Previous studies have also shown that β-catenin interacts with AR [85], which shares both structural and functional similarities with GR [12,13,86]. Consistent with these observations and our STRING analysis, we observed that LEDGF/p75 co-immunoprecipitated with β-catenin in 22Rv1-DR, PC3-DR, and DU145-DR cells and with AR in 22Rv1-DR cells (Appendix A). 

The subcellular localization of GR and LEDGF/p75 was assessed by highly sensitive confocal microscopy after treatment with 100nM dexamethasone for 30 min, which is sufficient to induce GR translocation in PCa cells [25,38]. As expected, untreated PC3 and PC3-DR cells cultured in charcoal-stripped FBS displayed LEDGF/p75 in the nucleus (green staining) but GR in the cytoplasm (red staining) (Figure 6D,E, top panels). However, after dexamethasone treatment, activated GR translocated into the nucleus, and the merged images displayed a yellow color indicative of co-localization between GR and LEDGF/p75 (Figure 6D,E, bottom panels).

Taken together, these results suggest that LEDGF/p75 and GR are part of a large transcriptional network in DTX-resistant PCa cells that also includes β-catenin and AR. Based on our previous report [38] and the STRING analysis (Figure 5), this network also includes other members of the LEDGF/p75 IBD interactome such as Menin, MLL, JPO2, PogZ, and IWS1, as well as interacting partners of these IBD interactors such as BRD4, MED1 and MYC. 

### 3.5. Selective GR Modulators Resensitize DTX-Resistant Cells to DTX

Recently, we used the SGRM CORT108297 to target GR in combination with the β-catenin inhibitor MSAB in DTX-resistant PCa cells, observing that this combination significantly enhanced cellular responses to DTX [25]. In the present study, we determined whether the co-targeting of GR and LEDGF/p75 had similar effects in DTX-resistant PCa cells. For these experiments, we used the novel SGRMs exicorilant (EXI) and relacorilant (RELA). Recent studies showed that when combined with the taxane drug paclitaxel, RELA promoted chemotherapy response in cultured OVACR5 ovarian cancer cells, MIA PaCa pancreatic tumor xenografts, and patients with metastatic pancreatic cancer, ovarian cancer, and other solid tumors [87,88]. To assess the cytotoxicity of EXI and RELA, we first treated DTX-sensitive PCa cells with increasing concentrations of these two SGRMs (0–10,000 nM) for 72 h in the presence or absence of DTX. A moderate but significant decrease in viability was observed only in PC3 and MDA-PCa-2b cells treated with 10,000 nM EXI (Figure 7A,D, green bars) and in MDA-PCa-2b cells treated with 10,000 nM RELA (Figure 7H, yellow bars) in the absence of DTX. Only results obtained at 0, 100, 1000, and 10,000 nM were shown since neither of the two SGRMs had effects on cell survival at 0.1, 1, and 10 nM. We also conducted combinatorial treatments in these DTX-sensitive cells with the SGRMs in the presence of 10 nM DTX (maintenance concentration for DTX-resistant cells). However, we were unable to detect increased responses to DTX because these cell lines exhibit extensive cell death in the presence of 10 nM DTX alone (0 nM EXI or RELA, Figure 7A–H, grey bars).

We then proceeded to treat the DTX-resistant PCa cells with these SGRMs in the absence and presence of 10 nM DTX (maintenance concentration for DTX-resistant cells). A significant decrease in cell viability was observed in PC3-DR and DU145-DR cells treated with 10,000 nM EXI alone compared to the untreated controls (Figure 8A,B). The decrease in survival was even more robust and statistically significant when the DTX-resistant cells were treated with 10,000 nM EXI in the presence of 10 nM DTX (Figure 8A,B). A significantly increased response to DTX was also observed in DU145-DR cells at 1000 nM EXI (Figure 8B). In 22Rv1-DR cells, EXI alone decreased cell viability starting at 100 nM when compared to untreated controls (Figure 8C). However, in the presence of 10 nM DTX, the decrease in cell viability was more robust and significant at 1000 and 10,000 nM compared to cells with no DTX treatment (Figure 8C). Treatment with RELA alone did not influence survival in any of the three DTX-resistant cell lines (Figure 8D–F); however, like EXI, 10,000 nM RELA combined with 10 nM DTX significantly decreased survival in the three cell lines compared to RELA alone (Figure 8D–F). 

To better characterize the cytotoxic effects of EXI and RELA in combination with DTX on DTX-resistant cells, we treated PC3-DR, DU145-DR, and 22Rv1-DR cells with 1 μM, 5 μM, or 10 μM EXI or RELA in combination with increasing concentrations of DTX. We chose these three concentrations of EXI or RELA based on our observations in Figure 7 showing decreased cell viability when these SGRMs were used at 10 μM. The IC_50_ of DTX was calculated in PC3-DR (79.8 nM), DU145-DR (32.6 nM), and 22Rv1-DR (84.0 nM) (Appendix A) as a first step in determining how DTX efficacy is altered in these drug-resistant cell lines in the presence of EXI or RELA. We observed that EXI, used at 1, 5, and 10 μM, enhanced DTX cytotoxicity in PC3-DR cells by decreasing the DTX IC_50_ values to 32.1, 6.1, and 2.9 nM, respectively (Figure 9A). Similar decreases in DTX IC_50_ values were observed in DU145-DR (Figure 9B) and 22Rv1-DR (Figure 9C) at 5 μM and 10 μM EXI. While 1 μM EXI had no effect on the DTX IC_50_ values in DU145-DR cells (Figure 9B), this concentration decreased the IC_50_ value to 28.0 nM in 22Rv1-DR cells (Figure 9C). RELA demonstrated comparable results in all three DTX-resistant cell lines (Figure 9D–F).

To determine whether treatment of DTX-resistant cells with EXI or RELA in combination with DTX influences apoptosis markers, we examined, via immunoblotting, the effects of these combinations on Bcl-2 and PARP protein expression. In a previous study, Kroon and colleagues [23] reported increased Bcl-2 expression in DTX-resistant PCa cells and observed that GR antagonism with RU-486 (mifepristone) reverses this upregulation. Consistent with these results, we observed the increased expression of Bcl-2 in PC3-DR cells compared to the sensitive PC3 cells (Figure 10A,B). This upregulation was reversed by treatment with 10 μM EXI or RELA in combination with 10 nM DTX. In agreement with the induction of apoptosis by these combinatorial treatments, we observed that the combination of EXI or RELA with DTX led to the extensive degradation of PARP, a caspase substrate, in the sensitive PC3 cells, as well as extensive cell death visualized by microscopy. This is in agreement with the dramatic loss of viability in these sensitive cells induced by 10 nM DTX alone, which is shown in Figure 7. Cleavage of PARP into its 86 kD apoptotic signature fragment (cPARP) was associated with extensive cell death and was detected in PC3-DR cells treated with EXI or RELA in combination with 10 nM DTX (Figure 10C,D). Apoptosis in these DTX-resistant cells was most likely driven by the combinatorial treatment since these cells are viable in the presence of 10 nM DTX alone (0 nM EXI or RELA) (Figure 8, grey bars) and exhibit minimal loss of viability when treated with 10 μM EXI or RELA in the absence of DTX (Figure 8, green and yellow bars). Taken together, these results reinforce our observations that the treatment of DTX-resistant PCa cells with EXI or RELA in combination with DTX resensitizes these cells to this chemotherapeutic drug. 

### 3.6. Targeting of GR and LEDGF/p75 Decreases Clonogenicity in PC3-DR Cells in the Presence of Docetaxel

To further explore the anti-cancer effects of EXI and RELA in combination with DTX in chemoresistant PCa cells, we performed colony formation assays in PC3-DR cells. We observed that 1 μM EXI or RELA did not influence clonogenicity in PC3-DR cells when compared to vehicle controls (VEH) in the absence of DTX (Figure 11A,B). However, in the presence of 10 nM DTX, there was a robust and significant decrease in colony formation in cells treated with 1 μM EXI or RELA (Figure 11A,B). Colony formation was also significantly and dramatically suppressed in PC3-DR cells treated with 5 μM (Figure 11C,D) and 10 μM (Figure 11E,F) of EXI or RELA. These results are consistent with the results shown in Figure 8 and Figure 9 indicating that EXI and RELA enhance the response to DTX in DTX-resistant cells. 

We then determined the effects of the dual targeting of GR and LEDGF/p75 on clonogenicity in the presence and absence of DTX. For this experiment, we treated PC3-DR cells with 1 μM EXI or RELA in combination with LEDGF/p75 depletion. Because of the lack of a specific LEDGF/p75 inhibitor, we decided to target this protein genetically using RNA interference. We demonstrated previously that the siRNA-mediated depletion of LEDGF/p75 decreases, although it does not completely abolish, clonogenicity capacity in DTX-resistant cells treated with increasing DTX concentrations [29,38]. Transient LEDGF/p75 depletion was performed in PC3-DR cells and confirmed by immunoblotting when compared to the SCR negative control (Figure 11G), which was indicative of successful LEDGF/p75 depletion at the time of clonogenic growth, and treatments with EXI and RELA were initiated. We used the 1 μM EXI or RELA concentration to better visualize the changes in clonogenicity since we observed that 5 μM drastically decreased clonogenicity (Figure 11C,D). We observed that in the absence of DTX, 1 μM EXI or RELA did not affect PC3-DR clonogenicity in the SCR control samples; however, there was a slight but significant colony decrease in the VEH and 1 μM EXI groups when LEDGF/p75 was silenced compared to the SCR controls (Figure 11H). Of note, no statistically significant change in colony formation was observed upon treatment with 1 μM RELA + LEDGF/p75 depletion in the absence of DTX. Strikingly, the number of colonies significantly and markedly decreased in PC3-DR cells depleted of LEDGF/p75 and treated with 1 μM EXI or 1 μM RELA in the presence of 10 nM DTX (Figure 11I). These results suggest that the combined targeting of GR and LEDGF/p75 leads to a strong response to DTX in taxane-resistant PCa cells.

Taken together, the results shown in Figure 8, Figure 9, Figure 10 and Figure 11 provide valuable insights into the potential use of the SGRMs EXI and RELA in combination with taxanes such as DTX in overcoming PCa chemoresistance. These results also suggest that using the combined GR and LEDGF/p75 antagonism may be an adjuvant therapeutic approach for patients with mCRPC that failed DTX chemotherapy. 

### 3.7. RNA-Seq Analysis of Genes Differentially Regulated after GR or LEDGF/p75 Knockdown in DTX-Resistant PCa Cells Reveals Unique and Overlapping Transcriptomes

To obtain initial insights into the mechanisms by which the GR–LEDGF/p75 axis promotes chemoresistance in PCa cells, we conducted an RNA-seq analysis of PC3-DR and DU145-DR cells with individual siRNA-mediated knockdowns of GR or LEDGF/p75. The efficiency of the knockdowns after 72 h was confirmed by immunoblotting (Appendix A). Principal Component Analysis (PCA) of the RNA-seq data demonstrated that the DTX-resistant cells were clearly separated from their respective controls based on their global transcriptome expression profiles (Appendix A). For instance, PC3-DR and DU145-DR cells with GR silencing (siGR) clustered together, but this clustering was separate from that of cells with LEDGF/p75 silencing (siLEDGF/p75) or cells transfected with scrambled siRNA controls (SCRs) (Appendix A). The separate clustering of the siLEDGF/p75 and siGR samples was confirmed by correlation plot analysis (Appendix A) and suggested differences in the GR and LEDGF/p75 transcriptomic profiles. Our RNA-seq data also revealed 970 differentially expressed genes (DEGs) after silencing LEDGF/p75 and 670 DEGs after silencing GR, with statistical significance, in DU145-DR and PC3-DR cells, respectively when compared to their SCR counterparts (Appendix A). There were 321 overlapping DEGs between the siGR and siLEDGF/p75 groups, suggesting a partial shared transcriptome. A heatmap of the top 500 DEGs also demonstrated the clustering of the siLEDGF/p75 and siGR groups based on transcriptome expression profiles (Figure 12A).

To validate our study, we compared our RNA-seq data for siGR with RNA-seq data generated by Arora et al. [12] in enzalutamide-resistant cells with GR upregulation. These investigators identified SPOCK1, SNAI2, and PEMPA1 as genes downregulated when GR is upregulated in the context of enzalutamide resistance. Consistent with this, we observed that upon silencing GR in the DTX-resistant cells, there was an upregulation of these three genes (Appendix A).

We further analyzed the top ten DEGs in the siGR, siLEDGF/p75, and overlap groups, which identified that MUC2, VIL1, and SNGH25 were commonly downregulated in all three groups (Figure 12B). A heatmap of the top 50 ranked DEGs in the overlap group is shown in Appendix A. Gene Set Enrichment Analysis (GSEA) identified multiple gene pathways that were significantly enriched in the overlapping DEGs upon the silencing of GR and LEDGF/p75 individually in DTX-resistant PC3-DR and DU145-DR cells. We also identified pathways that are unique to GR or LEDGF/p75. Representative overlap pathways with statistical significance (*p* < 0.001) are shown in Figure 12C and include “Apoptosis”, “Reactive Oxygen Species”, “Androgen Response”, and “G2M Checkpoint”. These gene pathways are associated with the regulation of prostate cancer cells’ death and survival decisions, proliferation, and response to therapy.

## 4. Discussion

The development of therapy cross-resistance is a major challenge in the sequential treatment with ARSI and taxanes in patients with advanced PCa [26]. Growing evidence supports the notion that prostate tumors have pre-existing or acquired mechanisms to develop resistance to a particular drug or therapy following the acquisition of resistance to a preceding therapy (e.g., ARSI) [26]. For instance, resistance to other ARSI drugs and to DTX, but not to CBZ, after acquired resistance to enzalutamide and abiraterone has been observed in mCRPC [89,90,91]. In addition, mCRPC cells selected for resistance to DTX have shown cross-resistance to the taxanes CBZ and paclitaxel [29]. 

These observations suggest the existence of common mechanisms underlying therapy cross-resistance in mCRPC. These may include, for instance, the reactivation by ARSI treatment of AR variants such as AR-V7, which has been shown to contribute to ARSI resistance and is induced by taxanes in mCRPC cells and tissues [11,25]. However, clinically, AR-V7 does not appear to be essential for taxane chemoresistance since no correlation was found between AR-V7 expression and response to taxanes in patients with PCa [92,93,94]. The Wnt/β-catenin pathway has also been implicated in PCa therapy cross-resistance given the reported contribution of β-catenin to both ARSI and taxane resistance [25,95,96,97,98,99,100]. Another mechanism of PCa therapy cross-resistance is the upregulation of GR resulting from AR inhibition. It is now well-documented that GR contributes to ARSI resistance by bypassing AR inhibition through its ability to transcriptionally upregulate both AR- and GR-target genes associated with PCa progression [12,13,14,17,18,19,101]. GR activation has also been implicated in PCa resistance to taxane chemotherapy [20,23,25]. Although the mechanisms underlying GR-mediated PCa chemoresistance still remain to be fully elucidated, there is evidence for the downregulation of the anti-apoptotic genes Bcl-xL and Bcl-2 upon GR antagonism, as well as for the GR-mediated upregulation of mono amine oxidase-A (MAO-A), a mitochondrial oxidoreductase, in patients treated with DTX [20,23]. Recently, we also showed that GR and β-catenin are upregulated and interact in DTX-resistant PCa cells and that their antagonism attenuates chemoresistance [25].

Previously, our group reported that glucocorticoids upregulate the expression of clusterin and LEDGF/p75, two oncoproteins implicated in cancer chemoresistance, in PCa cells [24]. This upregulation was reversed by blocking GR signaling with mifepristone. In the present study, we provide evidence for the GR-mediated upregulation of LEDGF/p75 in PCa cells by showing that: (1) GR silencing led to robust and significant LEDGF/p75 protein and transcript downregulation in a panel of both DTX-sensitive and DTX-resistant cells; (2) GR and LEDGF/p75 were concomitantly upregulated in enzalutamide-resistant LNCaP cells, and GR silencing in these cells led to the downregulation of LEDGF/p75 expression; and (3) our analysis of publicly available ChIP-seq data showed the enrichment of GR in the promoter region of *PSIP1*, the gene encoding LEDGF/p75, in two PCa cell lines (LNCaP-1F5 and VCaP) and two leukemia cell lines (696 and Nalm6) that express high levels of GR. We should emphasize that both the LNCaP-1F5 cell line and our LNCaP-ENZR cell line overexpress GR and, to some extent, could be considered equivalent. Interestingly, HRP2, a transcription-associated protein that shares similar domain organization, interacting partners, and functions with LEDGF/p75 [38,48,49,50,51,78,79], was not downregulated by GR silencing. Although LEDGF/p75 and HRP2 cooperate in HIV integration and replication, cancer cell survival, facilitating RNAPII transcription, and chemoresistance, their functions are not completely redundant [38,48,49,50,51,102]. Given the overlapping pro-survival functions of these two proteins, it is plausible that cancer cells may maintain independent mechanisms for their regulation. 

The observed GR-mediated upregulation of LEDGF/p75 in enzalutamide-resistant PCa cells raises the intriguing possibility that this transcription co-activator may also contribute to ARSI resistance. While the most widely accepted mode of action of enzalutamide is targeting AR activity, there is evidence that this drug also triggers apoptosis and attenuates the expression of anti-apoptotic proteins and heat shock proteins such as HSP27 in PCa cells and other cancer cell types [103,104,105,106]. Targeting anti-apoptotic and cell survival signaling pathways has been proposed as a novel strategy to overcome enzalutamide resistance in PCa [18,107,108,109]. LEDGF/p75 is a stress-induced pro-survival protein that protects cancer cells against both apoptotic and non-apoptotic cell death induced by anticancer drugs including taxanes and DNA-damaging agents [27,29,30,31,34,35,36,37,38,46,110]. These pro-survival functions have been linked to the ability of LEDGF/p75 to transactivate genes associated with stress and antioxidant responses, heat shock proteins including HSP27, cell cycle progression, angiogenesis, and Hox signaling [32,33,39,45,111,112,113]. The contribution of LEDGF/p75 to PCa enzalutamide resistance remains to be determined in pre-clinical models and clinical tumors. 

Our immunoprecipitation and confocal microscopy studies suggest that GR and LEDGF/p75 are part of a transcription complex associated with active chromatin in DTX-resistant cells, as indicated by the co-immunoprecipitation of the active chromatin marker H3K36me2. This is consistent with our previous studies showing that GR signaling is required for the nuclear translocation of the c-MYC interacting protein JPO2, which co-localized in active chromatin with GR and LEDGF/p75 [38]. The observation that LEDGF/p75 also co-immunoprecipitated with AR and β-catenin, both interacting partners of GR [25], in diverse PCa cell lines further validated its presence in a large transcriptional network that is highly relevant to PCa therapy resistance. However, the interaction of LEDGF/p75 with these GR interacting partners is most likely indirect, through other interacting partners, as suggested by the STRING protein–protein interaction network analysis. Nevertheless, we cannot rule out the possibility that the PWWP and IBD domains of LEDGF/p75, which serve as hubs for multiple protein interactions with chromatin-associated proteins and transcription factors [38,52,53], may be involved in direct binding to GR and GR-interacting partners in specific contexts. Further studies are needed to examine the extent of protein interactome overlap between GR and LEDGF/p75 and determine whether this overlap is driven by direct or indirect interactions. 

We observed that GR antagonism with EXI and RELA enhanced the sensitivity of chemoresistant PCa cells to DTX, consistent with our recent studies using a different SGRM (CORT108297) [25] and a previous report from another group showing that GR antagonism with mifepristone and cyproterone acetate reverts chemoresistance in DTX-resistant PCa cells [23]. Ongoing Phase 1 clinical trials (NCT03437941 and NCT03674814) are evaluating the efficacy of EXI or RELA in combination with enzalutamide in patients with mCRPC. Preliminary results on the safety, tolerability, pharmacokinetic (PK), and pharmacodynamic (PD) of EXI + enzalutamide were recently reported (https://www.annalsofoncology.org/article/S0923-7534(22)03734-6/fulltext, accessed on 8 June 2023). Pre-clinical studies also showed that RELA increased the response to the taxane nab-paclitaxel in ovarian and pancreatic tumor cells and xenografts [87,88]. A Phase 1 trial (NCT02762981) in patients with various solid tumors established the tolerability of RELA + nab-paclitaxel (https://ascopubs.org/doi/10.1200/JCO.2018.36.15_suppl.2554, accessed on 8 June 2023), whereas the Phase 2 study (NCT03776812) of this combination in patients with platinum-resistant ovarian cancer showed improved progression-free survival (PFS) and duration of response (DOR) (https://ascopubs.org/doi/abs/10.1200/JCO.2022.40.17_suppl.LBA5503, accessed on 8 June 2023). Additional clinical trials should be designed to investigate the combination of EXI and RELA with the taxanes DTX and CBZ in patients with mCRPC. Further, our observation that the sensitivity to DTX in chemoresistant PCa cells was significantly augmented when EXI and RELA were used in combination with LEDGF/p75 silencing suggests that combining LEDGF/p75 inhibitors with these SMGRs could be an effective therapeutic strategy to increase prostate tumor response to taxanes. Ongoing efforts in the discovery of small molecule inhibitors targeting the HIV integrase–LEDGF/p75 interaction [114,115] may potentially yield LEDGF/p75 inhibitors that could be repurposed for the treatment of advanced PCa in combination with SMGRs and taxanes. 

The hierarchal clustering heatmap generated from our RNA-seq studies showed that individualknockdowns of GR and LEDGF/p75 in DTX-resistant cells impacted 321 common DEGs, suggesting that these two proteins have overlapping transcriptomes. This is consistent with our findings that GR regulates and interacts with LEDGF/p75 and that their co-targeting resensitizes these cells to DTX. MUC2, VIL1, and SNGH25 were among the top genes downregulated in response to either GR or LEDGF/p75 depletion. These genes have been linked to therapy resistance and cancer progression in human tumors. For instance, MUC2 overexpression correlates with resistance to concurrent chemoradiotherapy in colorectal cancer and has been linked to poor prognosis [116,117]. VIL1 has been identified as a novel marker for poor response to radiation treatment for patients with cervical adenocarcinoma [118,119]. SNGH25, a long noncoding RNA, is associated with poor prognosis in various cancers such as glioblastoma, endometrial cancer, and prostate cancer [120,121,122]. 

GSEA analysis revealed DEG enrichment upon GR or LEDGF/p75 depletion in several cellular pathways associated with cancer cell survival and therapy resistance. For instance, the “Apoptosis” and “Reactive Oxygen Species” pathways are consistent with the stress survival functions of GR and LEDGF/p75. The “Androgen Response” pathway is linked to the previously reported inverse correlation between AR signaling and GR expression [12,13]. It should be noted that the PC3-DR and DU145-DR cell lines do not express AR but express elevated levels of GR [23,25]. As PCa becomes resistant to ARSI, overexpressed GR bypasses AR and takes over the regulation of certain AR-target genes [12,13]. Consistent with this, we observed the decreased expression of AR response genes in the “Androgen Response” enrichment plot. The “G2M Checkpoint” pathway was also enriched in our GSEA analysis, being indicative of the differential expression of cell cycle genes influenced by GR or LEDGF/p75 depletion. This is consistent with previous studies showing that LEDGF/p75 drives breast cancer tumorigenicity by promoting the transcription of cell cycle genes [39]. Ongoing studies from our group are (1) evaluating, in depth, the RNA-seq data for GR and LEDGF/p75 depletion in DTX-resistant PCa cells with the goals of validating and linking mechanistically specific target genes to PCa taxane resistance, (2) cross-comparing our RNA-seq data with other RNA-seq data sets derived from the silencing of GR, LEDGF/p75, and selected interacting partners in different cancer contexts to identify critical overlapping pathways, and (3) determining the association of LEDGF/p75 and selected target genes and interacting partners with prognosis and therapy responses in patients with advanced PCa.

## 5. Conclusions

The results presented here reveal the following novel observations: (1) GR silencing in a panel of DTX-resistant and DTX-sensitive PCa cells led to the downregulation of LEDGF/p75 expression but not to that of its paralog HRP2; (2) GR upregulation in enzalutamide-resistant PCa cells was associated with LEDGF/p75 upregulation, and the knockdown of GR in these cells resulted in LEDGF/p75 downregulation; (3) ChIP-seq analysis revealed the presence of GR binding sites in promoter regions of the LEDGF/p75 gene; (4) GR and LEDGF/p75 are part of a large transcriptional network that includes several transcription factors and regulators linked to therapy resistance in PCa and other cancer types; (5) the pharmacological targeting of GR with the novel SGMRs EXI and RELA significantly increased the response of chemoresistant PCa cells to DTX, and this response was further augmented when GR antagonism was combined with LEDGF/p75 silencing; and (6) RNA-seq analysis of DTX-resistant cells with GR or LEDGF/p75 depletion revealed both differentially regulated genes and pathways that are unique to each of these proteins as well as overlap pathways associated with cancer cell death and survival decisions and therapy resistance. Taken together, these results implicate the GR–LEDGF/p75 axis in PCa cell resistance to taxane therapy and are consistent with growing evidence demonstrating a role for these proteins in therapy resistance in various human cancer types. The GR-mediated upregulation of LEDGF/p75 in enzalutamide-resistant cells needs to be further evaluated in future mechanistic and clinical studies with patient samples aimed at determining the possible contribution of LEDGF/p75 and its interacting partners regulated by GR, such as the MYC-binding protein JPO2, to PCa resistance to enzalutamide and other ARSI. Such studies are likely to identify the GR–LEDGF/p75 transcriptional network as a novel driver of therapy cross-resistance and an attractive target for adjuvant therapies using SGRMs and LEDGF/p75 inhibitors that are designed to increase taxane response in PCa patients.

## Figures and Tables

**Figure 1 cells-12-02046-f001:**
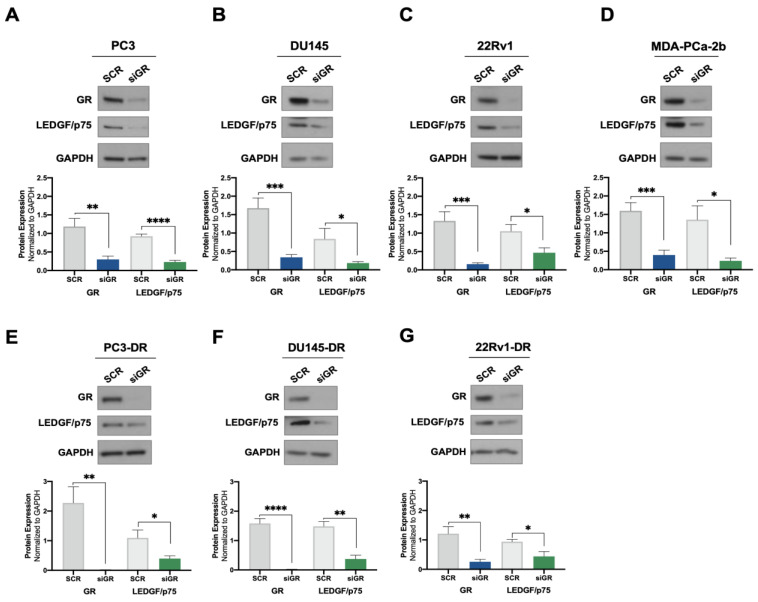
GR silencing leads to decreased LEDGF/p75 protein expression levels in PCa cells. DTX-sensitive PC3 (**A**), DU145 (**B**), 22Rv1 (**C**), and MDA-PCa-2b (**D**) cell lines were transfected with siRNAs specific for GR or scrambled negative control oligos (SCR) for 72 h. GR silencing resulting in diminished LEDGF/p75 expression was confirmed by immunoblotting. Similar results were observed in the DTX-resistant PCa cell lines PC3-DR (**E**), DU145-DR (**F**), and 22Rv1-DR (**G**). Quantified band values for GR and LEDGF/p75 were obtained with ImageJ software and plotted as relative protein expression normalized to GAPDH. Statistical analyses were performed using unpaired *t* tests comparing SCR to siGR samples. * *p* < 0.05, ** *p* <0.01, *** *p* < 0.001, **** *p* < 0.0001. Error bars represent means +/− SEM from at least 4 independent experiments for each cell line.

**Figure 2 cells-12-02046-f002:**
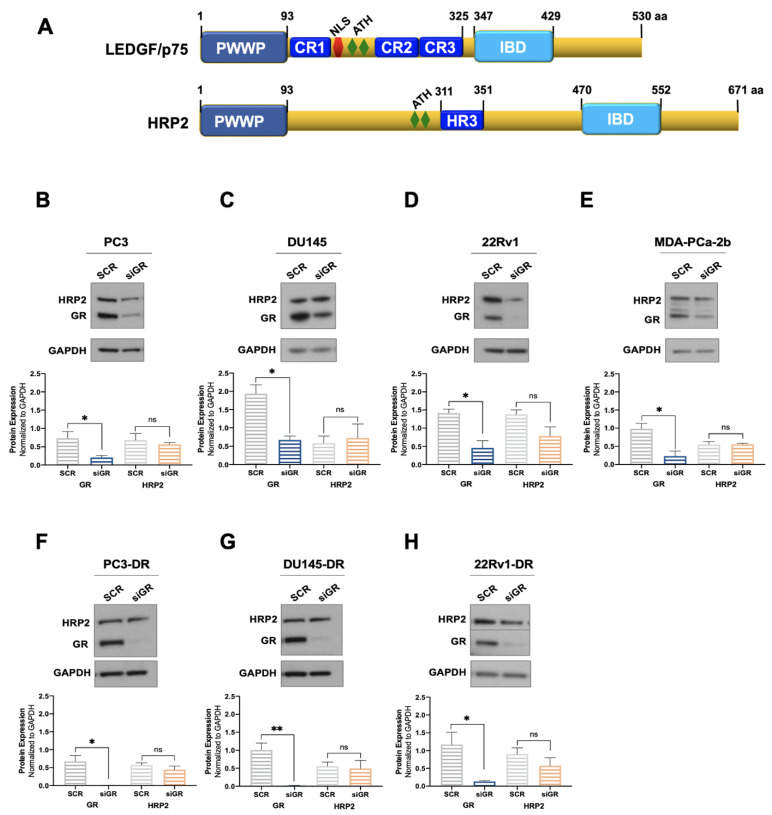
GR silencing has no effect on HRP2 protein expression in PCa cells. The domain structures of LEDGF/p75 and HRP2 share PWWP and IBD domains (**A**). LEDGF/p75 also has three charged regions (CR), a nuclear localization sequence (NLS), and AT-hook motifs. HRP2 has AT-hook motifs and a homology region III (HR3). DTX-sensitive PC3 (**B**), DU145 (**C**), 22Rv1 (**D**), and MDA-PCa-2b (**E**) cell lines were transfected with siRNAs specific for GR or scrambled negative control oligos (SCR) for 72 h. GR silencing associated with no significant change in HRP2 protein expression was confirmed by immunoblotting. Similar results were observed in the DTX-resistant PCa cell lines PC3-DR (**F**), DU145-DR (**G**), and 22Rv1-DR (**H**). The HRP2 and GR bands shown in panel H were from the same blot but had to be spliced due to the different exposure times used for their optimal detection. Quantified band values for GR and HRP2 were obtained with ImageJ software and plotted as relative protein expression normalized to GAPDH. Statistical analyses were performed using unpaired *t* tests comparing SCR to siGR samples. * *p* < 0.05, ** *p <* 0.01. Error bars represent means +/− SEM from 3 independent experiments for each cell line.

**Figure 3 cells-12-02046-f003:**
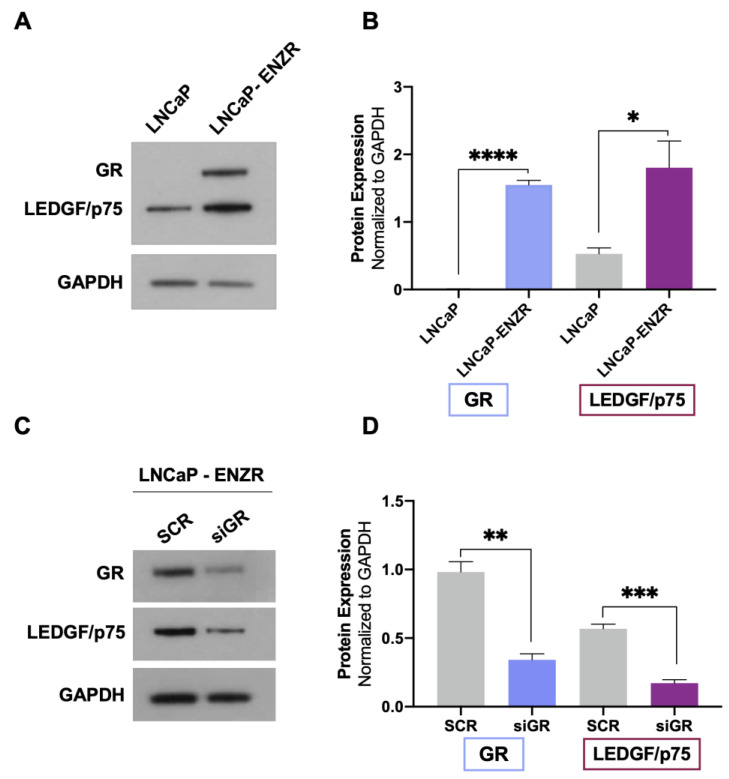
GR upregulation in LNCaP-ENZR cells correlates with increased LEDGF/p75 expression. The protein expression of GR and LEDGF/p75 was assessed by immunoblotting in LNCaP and LNCaP-ENZR cells (**A**). Transient GR silencing for 72 h in LNCaP-ENZR cells led to decreased LEDGF/p75 expression (**C**). Quantified band values for GR and LEDGF/p75 were obtained with ImageJ software and plotted as relative protein expression normalized to GAPDH (**B**,**D**). Statistical analyses were performed using unpaired *t* tests comparing SCR to siGR. * *p* < 0.05, ** *p <* 0.01, *** *p* < 0.001, **** *p* < 0.0001. Error bars represent means +/− SEM from 3 independent experiments.

**Figure 4 cells-12-02046-f004:**
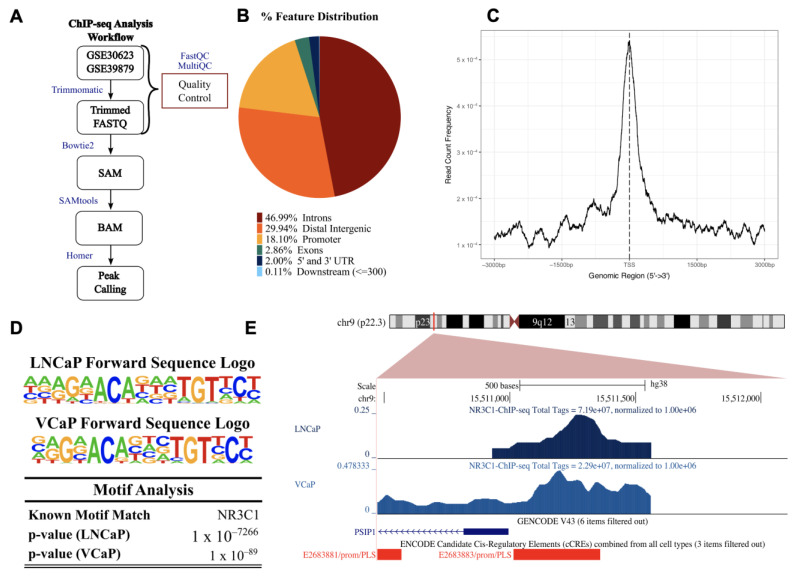
Chromatin immunoprecipitation-sequencing (ChIP-seq) for GR (NR3C1) in prostate cancer cell lines (datasets GSE30623 and GSE39879). (**A**) Flow diagram of the analysis pipeline used for the LNCaP-1F5 and VCaP ChIP samples obtained from the gene expression omnibus (GEO) database. (**B**) Binding sites were distributed among multiple features (breakdown of features: first intron = 14.87%, other introns = 32.12%, distal intergenic = 29.94%, 1–2 kb promoter = 4.65%, 2–3 kb promoter = 3.94%, <1 kb promoter = 9.51%, first exon = 0.65%, other exons = 2.21%, 5′ UTR = 0.16%, 3′ UTR = 1.84%, <300 bp downstream = 0.11%). (**C**) Average profile of ChIP peaks binding to transcription start site (TSS) regions (zero is set as the TSS). (**D**) Near the TSS of the *PSIP1* promoter region, HOMER motif analysis identified potential binding sites for NR3C1 in LNCaP-1F45 and VCaP. (**E**) UCSC human genome browser (GRCh38) visualization of normalized peaks for NR3C1 near the TSS of *PSIP1.* ChIP tracks are shown in shades of blue, GENCODE V43 *PSIP1* transcript is in blue, and ENCODE promoter-like signatures are in red.

**Figure 5 cells-12-02046-f005:**
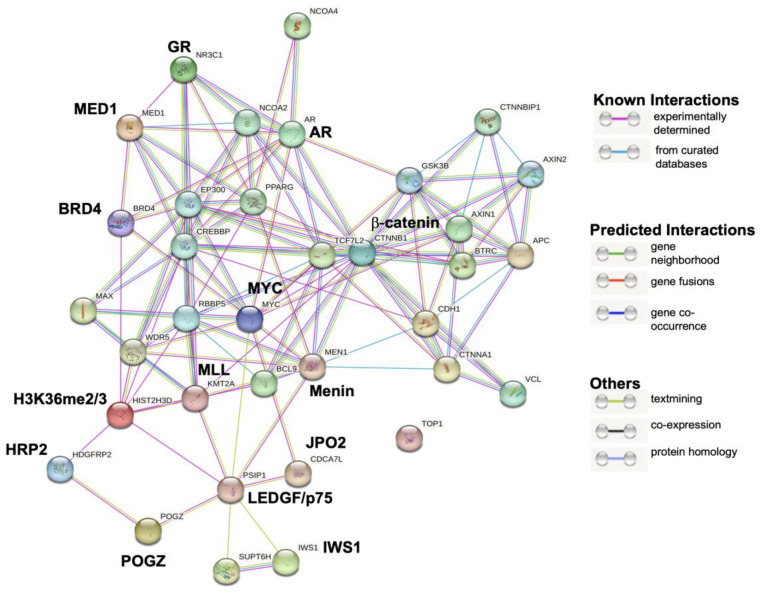
GR and LEDGF/p75 interaction module generated by STRING analysis. GR is encoded by the *NR3C1* gene and LEDGF/p75 by the *PSIP1* gene. The LEDGF/p75 interactions with GR, HRP2, β-catenin, and AR detected in this analysis appeared to be indirect as part of a common network.

**Figure 6 cells-12-02046-f006:**
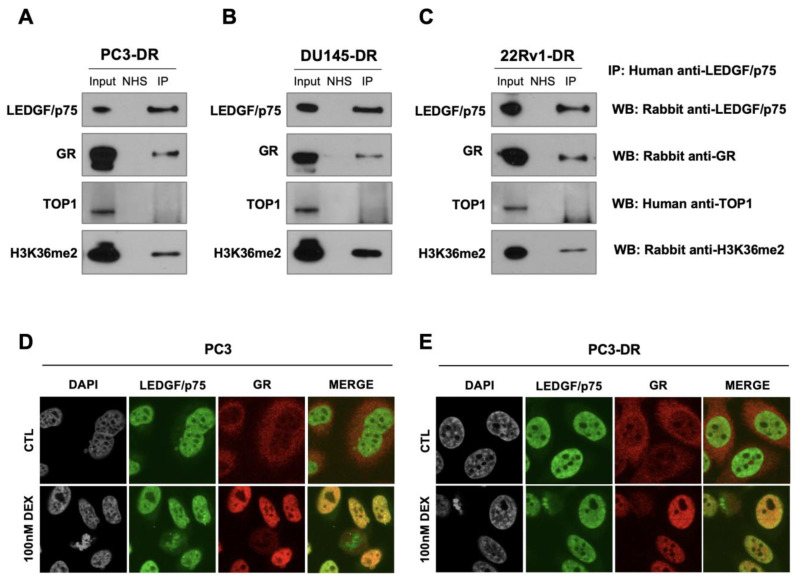
LEDGF/p75 and GR endogenously interact in DTX-resistant PCa cells in the presence of glucocorticoids. Immunoprecipitations were performed using specific human anti-LEDGF/p75 autoantibodies. IP was confirmed by immunoblotting with rabbit monoclonal antibodies specific for LEDGF/p75, GR, or H3K36me2 and human anti-TOP1 autoantibodies in PC3-DR (**A**), DU145-DR (**B**), and 22Rv1-DR (**C**). Normal human serum (NHS) was used as negative control for IP. Whole cell lysates collected from IP reactions were used as input (1% of IP). Cells were grown in normal culture medium, which contains glucocorticoids. Independent experiments were performed at least 3 times. Confocal microscopy analysis revealed LEDGF/p75 co-localization with activated GR in PC3 (**D**) and PC3-DR (**E**) cells after 30 min treatment with 100 nM dexamethasone (DEX) but not in control (CTL) cells with no DEX exposure. LEDGF/p75 displays the dense fine speckled nuclear pattern detected with FITC labeled secondary anti-human antibody (green). GR was detected with rhodamine-labeled secondary antibody (red). Merged images show yellow staining indicative of co-localization.

**Figure 7 cells-12-02046-f007:**
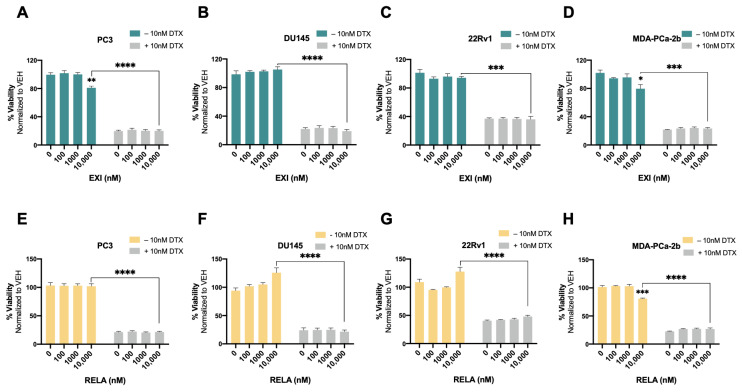
Selective GR modulators do not increase sensitivity to docetaxel in chemosensitive PCa cells. Cell viability was assessed using MTT assays after drug treatments for 72 h. DMSO was used as vehicle control. Exicorilant (EXI) dose response treatments were performed in the presence and absence of 10 nM DTX in PC3 (**A**), DU145 (**B**), 22Rv1 (**C**), and MDA-PCa-2b (**D**) cells. Similar results were observed with relacorilant (RELA) treatment in PC3 (**E**), DU145 (**F**), 22Rv1 (**G**) and MDA-PCa-2b (**H**) cells. Statistical analyses were performed using unpaired *t* tests. * *p* < 0.05, ** *p <* 0.01, *** *p* < 0.001, **** *p* < 0.0001. Error bars represent means +/− SEM from at least 3 independent experiments for each cell line.

**Figure 8 cells-12-02046-f008:**
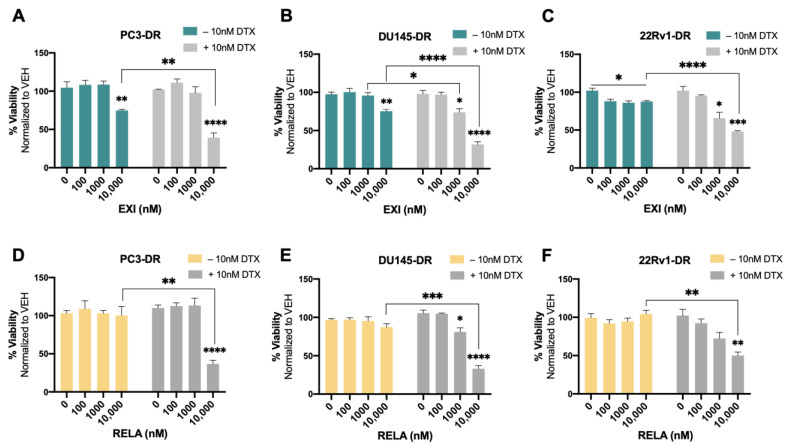
Selective GR modulators increase sensitivity to docetaxel in chemoresistant PCa cells. Cell viability was assessed using MTT assays after drug treatments for 72 h. DMSO was used as vehicle control. Exicorilant (EXI) dose response treatment was performed in the presence and absence of 10 nM DTX in PC3-DR (**A**), DU145-DR (**B**), and 22Rv1-DR (**C**) cells. DTX resensitization was observed at 10 μM EXI. Similar results were observed with relacorilant (RELA) treatment in PC3-DR (**D**), DU145-DR (**E**), and 22Rv1-DR (**F**) cells. Statistical analyses were performed using unpaired *t* tests. * *p* < 0.05, ** *p* < 0.01, *** *p* < 0.001, **** *p* < 0.0001. Error bars represent means +/− SEM from at least 3 independent experiments for each cell line.

**Figure 9 cells-12-02046-f009:**
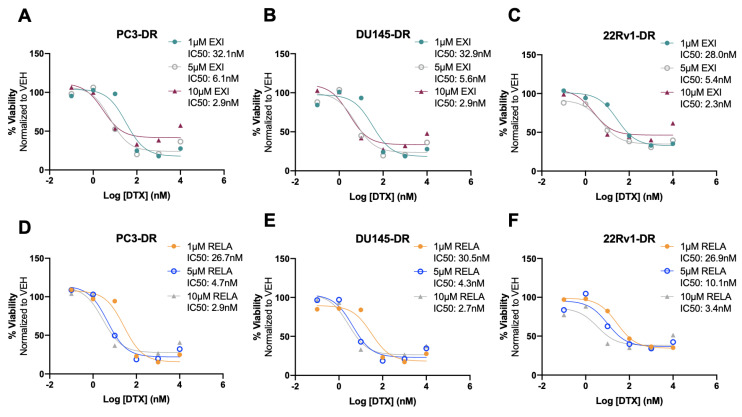
Selective GR modulators decrease docetaxel IC50 values in chemoresistant PCa cells. Cell viability was assessed using MTT assays after 72 h treatment of DTX-resistant cells with three different concentrations of exicorilant (EXI) and relacorilant (RELA) in the presence of increasing concentrations of DTX. DMSO was used as vehicle control. DTX dose response was performed in the presence of 1 μM, 5 μM, or 10 μM EXI in PC3-DR (**A**), DU145-DR (**B**), and 22Rv1-DR (**C**), or 1 μM, 5 μM, or 10 μM RELA in PC3-DR (**D**), DU145-DR (**E**), and 22Rv1-DR (**F**). Data represent at least 3 independent experiments for each cell line.

**Figure 10 cells-12-02046-f010:**
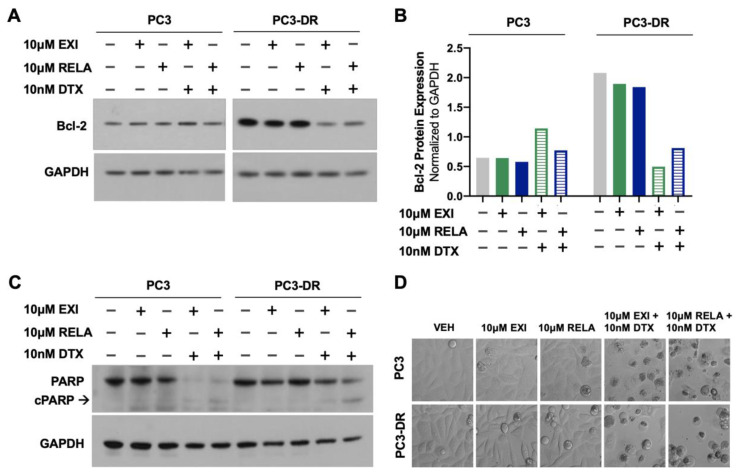
Selective GR modulators in combination with DTX decrease the expression of the anti-apoptotic protein Bcl-2 and increase PARP cleavage in DTX-resistant PCa cells. PC3 and PC3-DR cells were treated with 10 μM EXI, 10 μM RELA, 10 μM EXI + 10 nM DTX, or 10 μM RELA + 10 nM DTX for 72 h. Immunoblots show that PC3-DR cells had increased Bcl-2 protein expression levels compared to PC3 cells (**A**). Bcl-2 upregulation was reversed after treatment with EXI or RELA in combination with DTX but not by the GR modulators alone (**A**). The quantification of the blots from panel (**A**) is shown in panel (**B**). Extensive PARP degradation can be observed in PC3 cells after treatment with EXI or RELA in combination with DTX but not with the GR modulators alone (**C**). Apoptotic PARP cleavage (cPARP) can be detected in PC3-DR cells after treatment with EXI or RELA in combination with DTX but not with the GR modulators alone (**C**). Hoffman Modulation microscopy images show the cytotoxic effects of EXI and RELA, alone or in combination with DTX, in PC3 and PC3-DR cells (**D**).

**Figure 11 cells-12-02046-f011:**
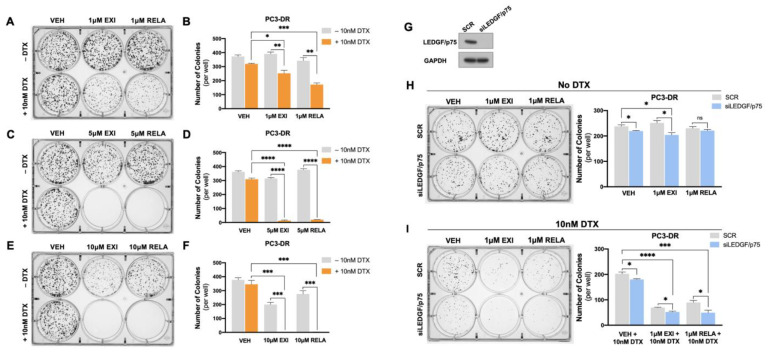
Targeting GR and LEDGF/p75 decreases colony formation capacity in PC3-DR cells in the presence of DTX. Representative images of PC3-DR colonies demonstrate a decrease in clonogenicity in the presence of 10 nM DTX with EXI or RELA concentrations of 1 μM (**A**,**B**), 5 μM (**C**,**D**), and 10 μM (**E**,**F**). For dual GR and LEDGF/p75 targeting, LEDGF/p75 knockdown in PC3-DR cells was confirmed by immunoblotting (**G**) at the time we initiated clonogenic growth and treatments with EXI and RELA, both in the absence (**H**) and presence (**I**) of DTX. Bar graphs show colony quantifications. Treatments with EXI and RELA were compared to their respective VEH controls. Colonies were analyzed after 10 days. Statistical analyses were performed using unpaired *t* tests. * *p* < 0.05, ** *p <* 0.01, *** *p* < 0.001, **** *p* < 0.0001. Error bars represent means +/− SEM from 3 independent experiments.

**Figure 12 cells-12-02046-f012:**
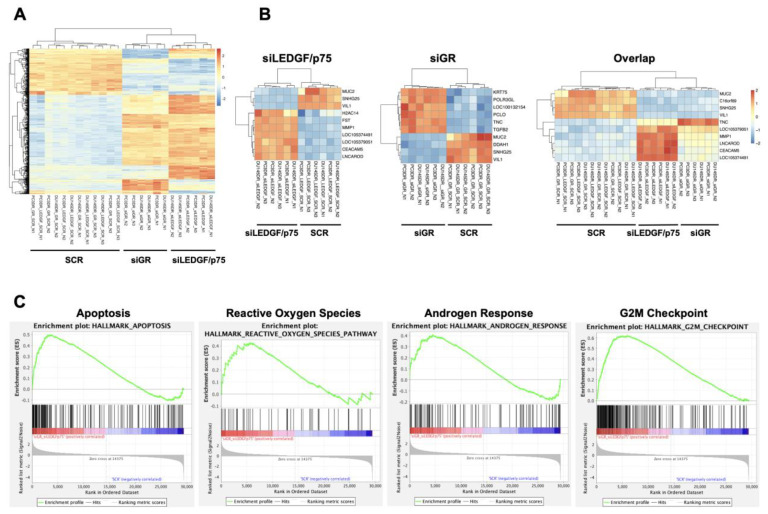
RNA-seq analysis of DTX-resistant PCa cells with GR or LEDGF/p75 depletion. (**A**) Hierarchical clustering heatmap of top 500 differentially expressed genes (DEGs) for SCR, siGR, and siLEDGF/p75 in DTX-resistant PCa cell lines (PC3-DR and DU145-DR). (**B**) Hierarchical clustering heatmap of top 10 DEGs for SCR vs siGR, SCR vs siLEDGF/p75, and overlapping SCR vs siGR and siLEDGF/p75 in PC3-DR and DU145-DR. (**C**) GSEA revealed four overlapping pathways (Apoptosis, Reactive Oxygen Species, Androgen Response, and G2M Checkpoint) that were significantly enriched upon individual GR and LEDGF/p75 silencing compared to scrambled siRNA controls (SCRs) (*p* < 0.001).

## Data Availability

The data presented here are available on request from the corresponding author.

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
