# Peer review of "Glucocorticoid Receptor Regulates and Interacts with LEDGF/p75 to Promote Docetaxel Resistance in Prostate Cancer Cells"

_cells, 2023, doi:10.3390/cells12162046_

Round 1
Reviewer 1 Report
Authors link glucocorticoid receptors to LEDGF/p75 in docetaxel resistant prostate cancer. This is an interesting study with potential translational implications. Some shortcomings ought to be addressed.
Comments
- Figure 1. Please show the viability (toxicity data) in Supplementary data.
- The relevance of the Chip seq data (3.3) is questionable since publicly available chipseq data of another prostate cancer cell line were used. It would have been nice if Chipseq data were obtained in the studied cell lines to check if GR binds to the promotor of LEDGF/p75
- Figure 6. Co-IP
- please include the parental cell lines that are DTX sensitive;
- was a DNAse treatment done to exclude that both proteins bind chromatin?
- Figure 7. Show the effect of the combination treatment (GR modulator + DTX) in DTX sensitive cell lines
- Show how the combination treatment (GR modulator + DTX) affects apoptosis and how GR or LEDGF/p75 regulates different apoptotic markers – this would increase the mechanistic understanding
Small Remarks
- How was the blotting done in Figure 2 with 2 distinct antibodies? Were the antibodies present together, was the blot cut and spliced, was re-stripping done? Please clarify. C and H show splicing…
- Typo line 571 (precipitate instead of precipitates)
no
Author Response
Response to Reviewer 1
Authors link glucocorticoid receptors to LEDGF/p75 in docetaxel resistant prostate cancer. This is an interesting study with potential translational implications. Some shortcomings ought to be addressed.
Comments
1. Figure 1. Please show the viability (toxicity data) in Supplementary data.
- In the new Supplementary Figure S1 we now show using Annexin V / 7-AAD staining followed by flow cytometry and calculation of apoptotic index, that GR silencing in PC3-DR and DU145-DR cell lines is not toxic to the cells when compared to the scrambled siRNA controls. These results are now addressed in Section 3.1, lines 357-361. The methods used are now presented in section 2.7, lines 250-259.
- As another control experiment to show that the reduction in GR and LEDGF/p75 expression in response to GR silencing is actually due to mRNA suppression and not to cellular toxicity, we conducted PCR studies in representative PCa cell lines and showed that GR silencing and the ensuing LEDGF/p75 downregulation occur at the transcript level. These data are now shown in the new Supplementary Figure S2, and addressed in section 3.1, lines 393-397.
2. The relevance of the Chip seq data (3.3) is questionable since publicly available chipseq data of another prostate cancer cell line were used. It would have been nice if Chipseq data were obtained in the studied cell lines to check if GR binds to the promotor of LEDGF/p75
-
- We agree with the reviewer that it would have been nice if the ChIP seq studies were done in house with data obtained in the cell lines being used in our laboratory. Unfortunately, it was not possible to conduct these studies in our laboratory during the manuscript revision period since they would take several weeks to complete. We should emphasize, however, that the publicly available ChiPseq data showing GR binding to the LEDGF/p75 promoter was confirmed not only in two PCa cell lines expressing GR (LNCaP-1F5 and VCAP) but also in two leukemia cell lines (697 and Nalm6) using data from two independently published studies. LEDGF/p75 is known to play an oncogenic role in both PCa and leukemia. In addition, we selected these published studies based on quantity of experimental replicates which had at least 3 independent replicates and the quality of antibody used for ChIP.
- We should also note that the publicly available ChiPseq data for PCa included an LNCaP cell line similar to the one used in our studies. This cell line has very low endogenous levels of GR as can be observed in Figure 3A. Because of this, the authors who generated the ChiPseq data used the LNCaP-1F5 cell line, which was engineered to express GR (Sahu et al., 2013). This LNCaP-1F5 model is to some extent equivalent to our LNCaP-ENZR model, which upregulates GR naturally upon development of enzalutamide resistance (Figure 3C). We showed that GR upregulation in LNCaP-ENZR cells correlates with LEDGF/p75 upregulation and that the latter can be reversed by GR silencing (Figure 3C,D). Further, as mentioned above, GR silencing in several PCa cell lines leads to LEDGF/p75 transcript downregulation (Supplementary Figure S2). These results suggest that GR regulates LEDGF/p75 expression, which is supported by the published ChIPseq data for two PCa cell lines and two leukemia cell lines. This issue is now discussed briefly in section 3.3., lines 490-494, and in the Discussion section, lines 859-869.
3. Figure 6. Co-IP, please include the parental cell lines that are DTX sensitive.
- We should mention that in our previous study on the LEDGF/p75 interactome in PCa (Ortiz-Hernandez et al, 2021) we focused our IPs only in the DTX-resistant cell lines because we were unable to attain efficient IP of LEDGF/p75 in the DTX-sensitive cell lines, which was likely due to the relatively low levels of LEDGF/p75 expression in these cell lines. To document this, we now include in the new Supplementary Figure S6 representative blots showing our attempts to IP LEDGF/p75 in the DTX-sensitive PC3 and DU145 cell lines. As can be observed, LEDGF/p75 could not be immunoprecipitated as we consistently demonstrated for the DTX-resistant cell lines. We concluded that LEDGF/p75 IP is inefficient in the sensitive cell lines, and perhaps could be only attained if larger numbers of DTX-sensitive cells are used in these experiments, compared to the numbers we normally use for the DTX-resistant cells. Having said this, confocal microscopy studies, which have superior sensitivity, conducted in DTX-sensitive PCa cells showed co-localization of GR and LEDGF/p75 in nuclei of cells treated with dexamethasone. This issue is now addressed in Section 3.4, lines 597-603.
4. Was a DNAse treatment done to exclude that both proteins bind chromatin?
-
- The commercial immunoprecipitation kit that we used for the IPs shown in Figure 6 neither contained DNAse nor indicated a DNase digestion step. However, as suggested by the reviewer, to rule out that the LEDGF/p75-GR interaction is dependent on the binding of both proteins to chromatin, we conducted a co-immunoprecipitation experiment in DU145-DR cells in the presence and absence of DNase treatment. As can be observed in the new Supplementary Figure S5, this treatment did not affect the co-immunoprecipitation of LEDGF/p75 and GR, suggesting that this interaction is mediated by protein-protein interactions rather binding of GR and LEDGF/p75 to chromatin. This is now addressed in Section 3.4, lines 591-597.
5. Figure 7. Show the effect of the combination treatment (GR modulator + DTX) in DTX sensitive cell lines
-
- We now provide a new Figure 7 that shows the effects of the combination treatment in four DTX sensitive cell lines. We were unable to detect increase responses to DTX because these cell lines exhibit extensive cell death in the presence of 10 nM DTX alone (see 0 nM EXI or RELA, Figure 7A-H). This is addressed in section 3.5, lines 634-645.
6. Show how the combination treatment (GR modulator + DTX) affects apoptosis and how GR or LEDGF/p75 regulates different apoptotic markers – this would increase the mechanistic understanding
- This is another great suggestion that we followed with additional experiments. In a previous study, Kroon et al (2016) reported increased Bcl-2 expression in DTX-resistant PC3 and DU145 cells compared to sensitive cells and observed that GR antagonism with RU-486 (mifepristone) reverses this upregulation. Consistent with these published findings, we observed increased expression of Bcl-2 in PC3-DR cells compared to the sensitive PC3 cells (new Figure 10A,B). This upregulation was reversed by treatment with 10mM EXI or RELA in combination with 10nM DTX. In agreement with the induction of apoptosis we also observed that the combination of EXI or RELA with DTX led to extensive degradation of PARP, a caspase substrate, in the sensitive PC3 cells as well as extensive cell death visualized by microscopy. This agrees with the dramatic loss of cell viability in these sensitive cells by 10nM DTX alone shown in Figure 7. Cleavage of PARP into its 86 kD apoptotic signature fragment (cPARP) was also associated with extensive cell death and was detected in PC3-DR cells treated with EXI or RELA in combination with 10nM DTX (new Figure 10C,D). Apoptosis in these DTX-resistant cells was likely driven by the combinatorial treatment since these cells are resistant to 10nM DTX alone (Figure 8) and exhibit minimal loss of viability in the presence of 10mM EXI or RELA in the absence of DTX (Figures 7 and 8). Taken together, these results indicate that treatment of DTX-resistant PCa cells with EXI or RELA in combination with DTX resensitizes these cells to this chemotherapeutic drug. This is now addressed in Section 3.5, lines 699-719.
- Regarding the question of how GR and LEDGF/p75 modulate apoptotic markers, our RNAseq data presented in Figure 12 shows that the independent silencing of GR and LEDGF/p75 in the DTX-resistant PC3-DR and DU145-DR cell lines yields several overlapping gene pathways that are influenced by both proteins. These include “Apoptosis”, “Reactive Oxygen Species”, “Androgen Signaling” and “G2M checkpoint”. All these pathways are associated with the regulation of cell death-survival decisions in PCa and other cancer cells. This is addressed in Section 3.7, lines 822-826. We plan to follow-up on this manuscript with additional mechanistic reports resulting from a more detailed mining and validation of the RNAseq data.
Small Remarks
- How was the blotting done in Figure 2 with 2 distinct antibodies? Were the antibodies present together, was the blot cut and spliced, was re-stripping done? Please clarify. C and H show splicing.
-
- We thank the reviewer for bringing to our attention this detail. In panel C the bands for HRP2 and GR came from the same blot using two distinct antibodies. In the original figure they were cut out and spliced, but in the revised figure are now shown in the same blot. In panel H the blot was first probed for GR, which migrates around 94 kD, to confirm the knockdown, and then separately probed with a different antibody for HRP2, which migrates around 120 kD. The bands, while coming from the same blot, had to be cut out and spliced because different exposure times were used for the optimal detection of antibody reactivity. This is now clarified in the legend of the revised Figure 2, lines 421-422. The original blots for panel C can be found in the Revised Original Western Blots page 6. The original blots for panel H can be found in the Revised Original Western Blots page 7.8.
- Typo line 571 (precipitate instead of precipitates). This has been corrected and appears now in line 586.

Reviewer 2 Report
In this study Sanchez-Hernandez et al shows that GR positively regulates expression of LEDGF/p75 in docetaxel (DTX)-resistant or sensitive PCa cells. This group further showed the GR up regulation in enzalutamide resistant LNCaP cells associated with p75 up regulation. Finally authors showed the co-targeting of both GR and LEDGF/p75 significantly increases the response of chemo resistant PCa cells to DTX. This is a significant study in principle to overcome the challenges of chemoresistance associated with various cancers.
Minor concerns.
1- Supplementary Figures are not available with this draft.
2- in figure 6 authors showed the interaction of GR and p75 in DEX resistant PCa cell lines what about these interaction in DEX sensitive PCa cell lines.
Since GR positively regulates P75 expression in both Dex sensitive and resistant cells equally it would be interesting to see whether there is a difference in the interactome of GR and P75 in DEX sensitive and resistant cells.
Author Response
Response to Reviewer 2
In this study Sanchez-Hernandez et al shows that GR positively regulates expression of LEDGF/p75 in docetaxel (DTX)-resistant or sensitive PCa cells. This group further showed the GR up regulation in enzalutamide resistant LNCaP cells associated with p75 up regulation. Finally, authors showed the co-targeting of both GR and LEDGF/p75 significantly increases the response of chemo resistant PCa cells to DTX. This is a significant study in principle to overcome the challenges of chemoresistance associated with various cancers.
Minor concerns.
- Supplementary Figures are not available with this draft.
- We regret that these supplementary figures were not available to the reviewer. These figures were uploaded during the original submission. It is possible that a glitch in the submission system prevented their proper uploading or viewing. The revised Supplementary Figure file has been uploaded with the re-submitted revised manuscript.
- In figure 6 authors showed the interaction of GR and p75 in DEX resistant PCa cell lines what about these interaction in DEX sensitive PCa cell lines.
- This important comment was also raised by Reviewer 1 (comment #3) and has already been addressed. New data shown in the new Supplementary Figure S6 and addressed in section 3.5, lines 597-603, demonstrates that we were unable to efficiently detect LEDGF/p75 IP in the DTX-sensitive cells.
- Since GR positively regulates P75 expression in both Dex sensitive and resistant cells equally it would be interesting to see whether there is a difference in the interactome of GR and P75 in DEX sensitive and resistant cells.
- This is a very important issue that we believe is out of the scope of this manuscript. As suggested by the STRING analysis (Fig. 5) and Supplementary Figure S7 we speculate that both proteins have common interacting partners such as MED1, AR and Beta-catenin. The question is if this common interactome is due to direct or indirect protein-protein interactions. We plan to conduct follow up studies to determine if the known LEDGF/p75 PWWP and IBD interactome (i.e MeCP2, JPO2, Menin, MLL, POGZ, etc) is also part of the GR interactome. We predict that some of these proteins will co-IP and co-localize with GR. These studies, however, will have to consider if GR interact with these LEDGF/p75 partners directly, which we don’t anticipate because of the lack of PWWP and IBD domains in GR, or indirectly through common interactors as suggested by the STRING analysis and Co-IP experiments. This is now addressed in the Discussion, lines 896-907.

Reviewer 3 Report
In this manuscript, entitled “Glucocorticoid Receptor Regulates and Interacts with LEDGF/p75 to Promote Docetaxel Resistance in Prostate Cancer Cells”, the authors evaluated the potential regulation of LEDGF/p75 by GR and that GR and LEDGF/p75 interact endogenously in the nucleus of DTX-resistant cells. Obviously, the authors put a lot of effort into this manuscript, some of the technical questions/suggestions are listed below:
1, In figue2D, I think there is a significant decrease of HRP2 in 22RV1.
2, The authors need to do ChIP-seq PCR to validate the results and use the Luciferase reporter gene detection system to prove the GR binding to the LEDGF/p75 promoter.
3, In Figure 6D, I can’t find the LEDGF/p75 and GR have the colocalization without dexamethasone treatment. So, the authors said” LEDGF/p75 and GR interact in DTX-resistant PCa cells” is inaccurate, they are not interacting with each other without dexamethasone treatment.
4, I didn’t find the supplementary Figure, and I think Supplementary Figure S4 are important figure, it shouldn’t be Supplementary Figure.
5, I don’t understand why the authors said: Glucocorticoid Receptor Regulates and Interacts with LEDGF/p75 to Promote Docetaxel Resistance in Prostate Cancer Cells. Does the LEDGF/p75 regulate the docetaxel resistance in prostate cancer cells? If the GR regulates the LEDGF/p75 expression and interact with the LEDGF/p75, KO GR or inhibit the GR function, it will inhibit the LEDGF/p75 function in mRNA and protein level, why combined targeting of GR and LEDGF/p75 leads to a strong response to DTX in taxane resistant PCa cells.
Author Response
Response to Reviewer 3
In this manuscript, entitled “Glucocorticoid Receptor Regulates and Interacts with LEDGF/p75 to Promote Docetaxel Resistance in Prostate Cancer Cells”, the authors evaluated the potential regulation of LEDGF/p75 by GR and that GR and LEDGF/p75 interact endogenously in the nucleus of DTX-resistant cells. Obviously, the authors put a lot of effort into this manuscript, some of the technical questions/suggestions are listed below:
- In fig. 2D, I think there is a significant decrease of HRP2 in 22RV1.
- We agree with the reviewer that there is definitively a noticeable reduction of HRP2 expression in 22Rv1 and 22Rv1-DR cells with GR silencing. However, this reduction did not achieve statistical significance (Fig. 2D,H). While these immunoblots were repeated independently three times, we cannot rule out that with additional replicates this reduction may attain significance. It is not clear, however, why GR would selectively influence HRP2 expression in 22Rv1 cell lines. This is now addressed in section 3.1, lines 410-415.
- The authors need to do ChIP-seq PCR to validate the results and use the Luciferase reporter gene detection system to prove the GR binding to the LEDGF/p75 promoter.
- We agree with the reviewer that ChIP-seq PCR and Luciferase Reporter assays would unambiguously prove that GR regulates the LEDGF/p75 gene. Based on our experience, these experiments will take some time to complete once we acquire all the reagents needed for the luciferase assays (kits, GR encoding plasmids, and LEDGF/p75 promoter plasmids). We felt that conducting these experiments would significantly delay the re-submission of the manuscript, especially given the relatively short time frame given to us by the publisher to re-submit the revised manuscript. We have already asked for two extensions to complete most of the experiments requested by the three reviewers. However, as indicated in the Discussion, lines 859-869, this manuscript provides solid evidence supporting a GR-mediated upregulation of LEDGF/p75 in PCa cells by showing that:
- GR silencing led to robust and significant LEDGF/p75 protein and transcript downregulation in a panel of both DTX-sensitive and DTX-resistant cells.
- GR and LEDGF/p75 were concomitantly upregulated in enzalutamide resistant LNCaP cells, and GR silencing in these cells led to downregulation of LEDGF/p75 expression.
- Analysis of publicly available ChIP-seq studies showed enrichment of GR in the promoter region of PSIP1, the gene encoding LEDGF/p75, in two PCa cell lines (LNCaP-1F5 and VCaP) and two leukemia cell lines (696 and Nalm6) that express high levels of GR. We should emphasize that both the LNCaP-1F5 cell line and our LNCaP-ENZR cell line overexpress GR and to some extent could be considered equivalent.
We plan to provide direct proof of the GR-mediated LEDGF/p75 regulation, with additional mechanistic data including regulatory elements, as part of our follow-up studies on the mining and validation of the GR and LEDGF/p75 overlap gene pathways from the RNAseq data.
- In Figure 6D, I can’t find the LEDGF/p75 and GR have the colocalization without dexamethasone treatment. So, the authors said” LEDGF/p75 and GR interact in DTX-resistant PCa cells” is inaccurate, they are not interacting with each other without dexamethasone treatment.
- In agreement with the reviewer, LEDGF/p75 would not be expected to interact nor colocalize with inactive GR, which is normally present in the cytoplasm in cells cultured in medium with charcoal-stripped FBS in the absence of Dexamethasone (DEX) (Fig. 6D,E, top control [CTL] panel). However, cells grown in normal culture medium, which is known to contain glucocorticoids (mentioned in line 567), or in medium with charcoal-stripped FBS in the presence of DEX show LEDGF/p75-GR colocalization (Fig. 6D,E, DEX panel). The IPs shown in Fig. 6A-C were conducted in cells grown in normal culture medium. Following the reviewer’s suggestion, we have modified the title and legend of Fig. 6 to clarify this issue (lines 574, 578, 580, 581)
- I didn’t find the supplementary Figure, and I think Supplementary Figure S4 are important figure, it shouldn’t be Supplementary Figure.
- The original Supplementary Figure S4 has been moved to the main manuscript text and is now the new Figure 7.
- As we indicated in our response to comment #1 of Reviewer 2, we regret that these supplementary figures were not available to the reviewer. These figures were uploaded during the original submission. It is possible that a glitch in the submission system prevented their proper uploading. The revised Supplementary Figure file has been uploaded with the re-submitted revised manuscript.
- I don’t understand why the authors said: Glucocorticoid Receptor Regulates and Interacts with LEDGF/p75 to Promote Docetaxel Resistance in Prostate Cancer Cells. Does the LEDGF/p75 regulate the docetaxel resistance in prostate cancer cells? If the GR regulates the LEDGF/p75 expression and interact with the LEDGF/p75, KO GR or inhibit the GR function, it will inhibit the LEDGF/p75 function in mRNA and protein level why combined targeting of GR and LEDGF/p75 leads to a strong response to DTX in taxane resistant PCa cells.
- This is a very good point that needs clarification. While we provide evidence that GR regulates LEDGF/p75 in PCa cells and that their dual targeting strongly sensitizes chemoresistant cells to DTX, we need to keep in mind that LEDGF/p75 is not the only protein regulating DTX resistance in PCa cells, and that GR is not the only transcription factor regulating LEDGF/p75 in cancer cells. DTX resistance is regulated by a large number of proteins including Bcl-2, HSP27, Clusterin, MDR1, beta-catenin, c-MYC, etc. LEDGF/p75 is known to be regulated by transcription factors AR and SP1 and other proteins. As shown in Figures 1, S2, and 3, a very efficient knockdown of GR not always leads to total LEDGF/p75 downregulation. In addition, as shown in Figure 3, LNCAP cells do not express GR but still express low levels of LEDGF/p75, consistent with other factors such as AR regulating the latter. While we believe that these two proteins act in concert to promote DTX resistance in PCa cells, and that GR regulates LEDGF/p75 in these cells, we can’t rule out that they may also contribute to DTX resistance through interactions with other transcription factors and chromatin proteins.

Round 2
Reviewer 1 Report
My comments were adequately addressed which I really appreciate.
Reviewer 3 Report
Agree to publish.